# Global endometrial DNA methylation analysis reveals insights into mQTL regulation and associated endometriosis disease risk and endometrial function

Sally Mortlock [1,21✉], Sahar Houshdaran [2,21], Idit Kosti[3,21], Nilufer Rahmioglu [4,5,21], Camran Nezhat[6,7,8], Allison F. Vitonis [9], Shan V. Andrews[3], Parker Grosjean[3], Manish Paranjpe[3], Andrew W. Horne [10], Alison Jacoby[2], Jeannette Lager[2], Jessica Opoku-Anane[2], Kim Chi Vo[2], Evelina Manvelyan[2], Sushmita Sen[2], Zhanna Ghukasyan[2], Frances Collins[10], Xavier Santamaria[11,12], Philippa Saunders [13], Kord Kober [3,14], Allan F. McRae[1], Kathryn L. Terry[9,15,16], Júlia Vallvé-Juanico [2,12], Christian Becker [5], Peter A. W. Rogers [17], Juan C. Irwin[2], Krina Zondervan [4,5,22], Grant W. Montgomery [1,22], Stacey Missmer[15,16,18,19,22], Marina Sirota [3,20,22] & Linda Giudice [2,22✉]

Endometriosis is a leading cause of pain and infertility affecting millions of women globally. Herein, we characterize variation in DNA methylation (DNAm) and its association with menstrual cycle phase, endometriosis, and genetic variants through analysis of genotype data and methylation in endometrial samples from 984 deeply-phenotyped participants. We estimate that 15.4% of the variation in endometriosis is captured by DNAm and identify significant differences in DNAm profiles associated with stage III/IV endometriosis, endometriosis sub-phenotypes and menstrual cycle phase, including opening of the window for embryo implantation. Menstrual cycle phase was a major source of DNAm variation suggesting cellular and hormonally-driven changes across the cycle can regulate genes and pathways responsible for endometrial physiology and function. DNAm quantitative trait locus (mQTL) analysis identified 118,185 independent *cis*-mQTLs including 51 associated with risk of endometriosis, highlighting candidate genes contributing to disease risk. Our work provides functional evidence for epigenetic targets contributing to endometriosis risk and pathogenesis. Data generated serve as a valuable resource for understanding tissue-specific effects of methylation on endometrial biology in health and disease.

A full list of author affiliations appears at the end of the paper.

Endometriosis is a common, estrogen-dependent, inflammatory disorder defined by the presence of endometrial-like tissue at extra-uterine (ectopic) sites, commonly within the pelvis[1]. It is associated with pain and infertility, affects an estimates 190 million women and contributes hundreds of billions of dollars to healthcare costs globally[1–4]. Pelvic disease likely derives from retrograde menstruation of steroid hormone-sensitive endometrial cells that elicit inflammation, neuroangiogenesis, fibrosis, and scarring[5–7]. Disease classification is based primarily on lesion appearance and location without correlation to symptoms or responses to surgical and hormonal therapies, and molecular classifications of disease subtypes are wanting. Although mechanisms driving the pathogenesis and pathophysiology of this heterogeneous disorder are largely unknown, studies suggest that diverse endometrial cellular components of women with disease respond abnormally to cyclic steroid hormones and exhibit aberrant signaling pathways, gene expression, decreased apoptosis, enhanced proliferation, and an inflammatory phenotype as compared to controls[8,9]. A recent genome-wide association meta-analysis identified disease-associated genomic loci in women with versus without endometriosis involving sex steroid and hormone signaling pathways, inflammatory pathways, oncogenesis and angiogenesis, with greater effect sizes observed in those with more advanced disease stage[10]. Genetic risk factors for endometriosis have also been associated with other reproductive traits including irregular menses, uterine fibroids, and ovarian cancer[10,11].

While genetic variation underpinning complex disease etiology is actively studied, epigenetic processes such as acquired alterations in DNA methylation (DNAm) are increasingly recognized as providing an important biological link between individual exposures and disease-specific phenotypes[12]. Endometrial whole methylome and candidate gene DNAm profiles in bulk tissue and select cell populations identify genes involved with steroid hormone dependence and abnormalities in women with versus without endometriosis. However, most results fail to be replicated due to limited sample size, cellular heterogeneity in bulk tissue specimens, poor cycle phase assignments, and limited clinical metadata[13–18]. Notably, aberrant candidate gene promoter DNAm includes HOX-A10[19], steroid hormone receptors (PR and ESR1)[13,20], CYP19/aromatase[14], SF-1[15,16], and COX-2[17] and altered expression of DNMTs[18]. DNA methylome-wide studies that use a small number of samples have suggested DNAm profiles differ in patients with endometriosis—both between endometriosis lesion and eutopic endometrial tissues, and compared to eutopic endometrium of women without disease[8,21–23]. One study of 16 patients with and without endometriosis was designed not to test the hypothesis of differential DNAm between cases and controls but to quantify intratissue variation and determine sample sizes necessary for large-scale discovery;[21] a second study evaluated the DNAm in 17 patients with endometriosis and 16 controls, across the menstrual cycle and noted marked phase- and disease-dependent signatures[9]. Both studies determined that menstrual cycle phase accounted for the majority of variability in DNAm patterns within the endometrium. Estradiol, progesterone, and their combination were found to regulate genome-wide DNAm signatures and gene transcription in endometrial stromal fibroblasts isolated from eight women with endometriosis (four with stage I/II and four with stage III/IV disease) and seven women without endometriosis, with marked aberrances in endometriosis patients[8]. Other recent small studies on genetic regulation of endometrial gene expression (expression quantitative trait loci; eQTL)[24] and DNAm (methylation quantitative trait loci; mQTL)[25] demonstrate associations between specific genotypes and expression of genes in signaling pathways and between

DNAm sites near GREB1 and KDR, genes involved in endometriosis pathogenesis.

This study reports global DNAm profiles and networks in endometrium associated with endometriosis, menstrual cycle and genetic variation using data collected from a large sample of patients with surgically and histologically confirmed endometriosis ($n = 637$) and women without endometriosis ($n = 347$). Importantly, the dataset is based on patients with well-documented clinical annotation, demographic characteristics, and symptomatology[26–28]. Our robust analysis reveals factors affecting epigenetic regulation in endometrium associated with endometriosis risk and disease heterogeneity and provides an important data resource for reproductive medicine.

## Results

**Sample characteristics**. Eutopic endometrial samples from 1074 women were included. Following sample quality control (QC) filtering, 984 participants (one sample each from 637 endometriosis cases, 347 controls (201 No Uterine or Pelvic Pathology (NUPP) controls)) contributed to the analyses (Table 1; Supplementary Data 1). The overall study design is shown in Supplementary Fig. 1. Clinical and demographic factors characterized across samples were: contributing site/institution, menstrual cycle phase, endometriosis case:control status, sample methylation array plate, sample methylation assay batch, revised American Society for Reproductive Medicine (rASRM) endometriosis disease stage[29], lesion type and pain variables (see Methods).

**Methylation in endometrium captures variation in endometriosis**. Using residual maximum likelihood analyses we estimated the proportion of variation in endometriosis case-control status captured by all genome-wide DNAm sites and compared this to the variation captured by genetics (Supplementary Data 2). Estimates of the variance captured by common SNPs (26.2% on the liability scale) was consistent with previously reported SNP-based heritability estimates[30]. The variance captured by SNPs can be interpreted as associated through causality however, variance captured by DNAm can reflect both causes and consequences of disease and therefore is dependent on the proportion of cases in the sample. As such, estimates for DNAm are read on the observed scale. We estimated that 24.2% of the variance was captured by all genome-wide DNAm sites. When including the genetic relationship matrix in the model, to account for variance captured by SNPs, the amount of variance explained by DNAm dropped to 16.1%. In total 37% of the variance in endometriosis case-control status was captured by a combination of both common genetic variants (20.9%) and DNAm in endometrium (16.1%).

**Single site and regional analysis identify distinct menstrual cycle phase DNAm profiles**. To investigate if variation in DNAm in endometrium was associated with menstrual cycle stage or endometriosis, single site (each individual DNAm site/probe) and regional association analyses were performed using a set of linear models. Genome-wide DNAm was measured across 759,345 DNAm sites in 984 samples using the Illumina Infinium MethylationEPIC Beadchip (Supplementary Fig. 1, Table 1). Genetic ancestry was determined using genotype data and principal component analysis (PCA); most of the participants were of European ancestry (68%) (Table 1, Supplementary Fig. 2a). PC-PR2 was used to estimate the amount of total variability of DNA methylation explained by technical covariates (institute, plate, batch) and sample characteristics (menstrual cycle phase, disease status, genetic ancestry). The largest contribution to the variability came from institute, cycle phase and batch explaining

**Table 1 DNAm sample information.**

| Variable | Classification | Differential DNAm analysis | | | | mQTL analysis | | | |
|---|---|---|---|---|---|---|---|---|---|
| | | Control | | Endometriosis | | Control | | Endometriosis | |
| | | n | % | n | % | n | % | n | % |
| Institute | University of Edinburgh | 31 | 9% | 52 | 8% | 29 | 13% | 50 | 11% |
| | University of Melbourne | 87 | 25% | 225 | 35% | 64 | 29% | 163 | 37% |
| | University of Oxford | 52 | 15% | 126 | 20% | 43 | 19% | 103 | 24% |
| | University of California San Francisco | 177 | 51% | 234 | 37% | 85 | 39% | 121 | 28% |
| Cycle phase | Proliferative | 170 | 49% | 303 | 48% | 98 | 44% | 198 | 45% |
| | Secretory (undefined sub-phase) | 7 | 2% | 15 | 2% | 3 | 1% | 9 | 2% |
| | Early Secretory | 44 | 13% | 78 | 12% | 35 | 16% | 57 | 13% |
| | Mid Secretory | 78 | 22% | 131 | 21% | 52 | 24% | 96 | 22% |
| | Late Secretory | 35 | 10% | 73 | 11% | 22 | 10% | 50 | 11% |
| | Menstrual | 13 | 4% | 37 | 6% | 11 | 5% | 27 | 6% |
| Genetic ancestry | Admixed | 27 | 8% | 55 | 9% | n/a | n/a | n/a | n/a |
| | African | 35 | 10% | 13 | 2% | n/a | n/a | n/a | n/a |
| | Admixed American | 22 | 6% | 30 | 5% | n/a | n/a | n/a | n/a |
| | East Asian | 28 | 8% | 50 | 8% | n/a | n/a | n/a | n/a |
| | European | 226 | 65% | 451 | 71% | 221 | 100% | 437 | 100% |
| | South Asian | 9 | 2% | 38 | 6% | n/a | n/a | n/a | n/a |
| Disease stage | I–II | n/a | n/a | 344 | 54% | n/a | n/a | 261 | 60% |
| | III–IV | n/a | n/a | 286 | 45% | n/a | n/a | 171 | 39% |
| | Unknown | n/a | n/a | 7 | 1% | n/a | n/a | 5 | 1% |
| Additional characteristics | Age (Mean ± SD) | 37.0 ± 8.4 (n = 342) | | 34.2 ± 7.1 (n = 634) | | 36.1 ± 8.6 (n = 220) | | 33.7 ± 7.1 (n = 436) | |
| | BMI (Mean ± SD) | 26.6 ± 6.1 (n = 301) | | 24.8 ± 5.0 (n = 591) | | 26.4 ± 6.1 (n = 196) | | 24.9 ± 5.1 (n = 414) | |
| | Parity (Mean ± SD) | 1.0 ± 1.35 (n = 325) | | 0.6 ± 1.0 (n = 613) | | 0.9 ± 1.4 (n = 206) | | 0.5 ± 1.0 (n = 422) | |
| | Age at menarche (Mean ± SD) | 12.8 ± 1.6 (n = 146) | | 12.7 ± 1.7 (n = 486) | | 12.9 ± 1.6 (n = 105) | | 12.8 ± 1.6 (n = 339) | |

The number and percentage of samples included in the DNA methylation (DNAm) analysis from each institute, at each cycle phase, from each genetic ancestry and with each disease stage.

43.53%, 2.99% and 1.43% of overall methylation variation, respectively (Supplementary Data 3). Following correction using surrogate variable analysis (SVA), institute explained only 0.53% whilst cycle phase and endometriosis status, which were protected for, explained 4.30% and 0.03%, respectively. DNAm data for all 984 samples before and after correction for covariates using SVA are shown in Supplementary Figs. 2b, c, 3. Including the surrogate variables (SVs) in the linear models reduced the association of the PCs with batch and site and identified a strong association with menstrual cycle phase (Supplementary Fig. 4). The largest number of differentially methylated sites was observed between samples collected within the secretory phase versus the proliferative phase (9,654 DNAm sites) and subsequent comparisons involving the aggregated phases (Fig. 1a, b, Supplementary Figs. 5, 6, Table 2, Supplementary Data 3, 4) regardless of disease status. Subtle differences were also observed across secretory (progesterone-dominant) sub-phase (early (ESE), mid (MSE) and late (LSE)) comparisons (Supplementary Data 4, 5). Clear clustering across cycle phase groups was observed (Fig. 1 and Supplementary Figs. 5, 6). The inflation lambda factors and the QQ plots are shown in Supplementary Fig. 7. The distribution of the genomic locations of the significantly differentially methylated sites is shown in Supplementary Data 6. Genes annotated to DNAm sites with differential DNAm between PE and SE were enriched in extracellular matrix (ECM)-cell interaction (adherens junctions, focal adhesion, regulation of actin cytoskeleton, Rho and Rap1 signaling) and cell proliferation and metabolism signaling (phospholipase D, PI3K-Akt, Ras signaling) pathways, consistent with known biological processes in the estrogen-dominant PE and progesterone-dominant SE[31] (Fig. 1c; Supplementary

Data 7). Several pathways were also enriched for genes annotated to significant DNAm sites from the stratified analysis which included ESE, MSE, and LSE phases, these included pathways unique to the ESE vs MSE comparison (e.g. CRMPs in Sema3A signaling) and ESE vs LSE comparison (e.g. apoptosis; Supplementary Data 7). Results from the differential methylation analysis using this SVA approach were consistent with an alternate Mixed Linear Model-based Omic Association (MOA) approach (Supplementary Fig. 8).

**Evidence of methylation differences in severe endometriosis.** Overall, we observed no significant difference (at genome-wide threshold) in single site or regional methylation signals between cases and controls. As effect sizes for many genetic risk factors are consistently greater in patients with rASRM stage III/IV disease[32], we analyzed results for stage III/IV cases compared with all controls. Two signals (cg02623400 in the 5'UTR region of *ELAVL4* on chromosome 1 and cg02011723 in the promoter region of *TNPO2* on chromosome 19) had higher DNAm in stage III/IV cases and passed the threshold for Bonferroni-corrected significance for a single test ($P < 6.58 \times 10^{-8}$). We also conducted an analysis of cases compared to the restricted NUPP control group. Only one site (cg26912870; intergenic) with lower DNAm on chromosome 11 in cases, and one site on chromosome 19 (cg18305031; intron of *EEF2*) with higher DNAm in stage III/IV cases, passed the threshold for Bonferroni-corrected significance for a single test. Differences in DNAm at cg02623400 and cg0201172 were only nominally significant when restricted to NUPP controls.

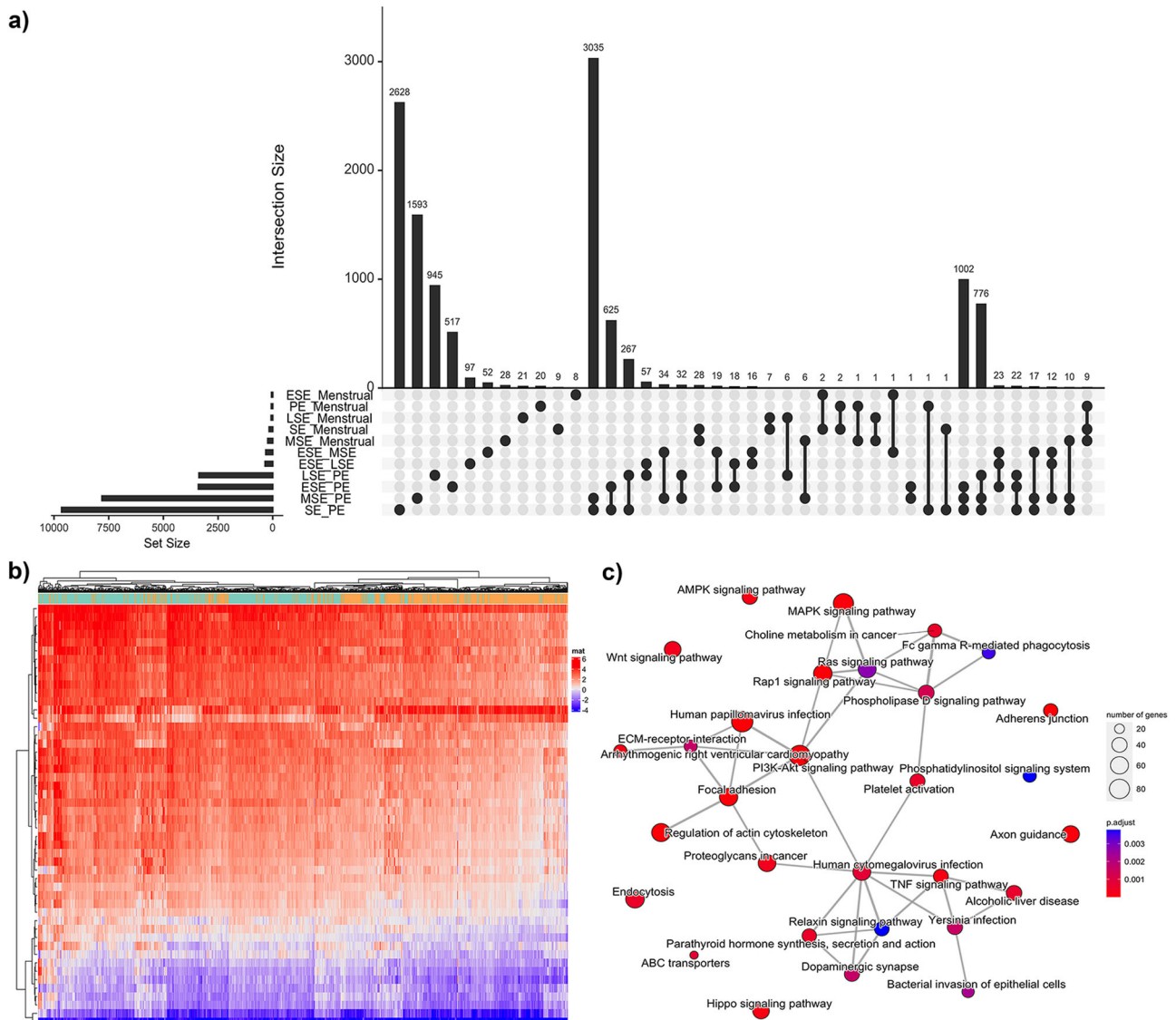

**Fig. 1 Cycle phase differences. a** Upset plot showing the number of significantly differently methylated sites between each comparison on the *y*-axis bar plot and the number of intersecting DNA methylation (DNAm) sites on the *x*-axis bar plot. The comparisons included in the intersection are depicted by dots and lines adjoining those dots. A total of 984 samples were included in these analyses (menstrual *n* = 50, proliferative (PE) *n* = 473, secretory (SE) *n* = 461, early secretory (ESE) *n* = 122, mid secretory (MSE) *n* = 209, late secretory (LSE) *n* = 108). **b** Heatmap of the top 50 most differentially methylated sites between the secretory (orange) and proliferative phase (green) of the menstrual cycle as annotated on the bar above the heatmap. Red denotes positive methylation M-values and blue denotes negative methylation M-values. **c** Pathways significantly enriched for genes annotated to differentially methylated sites between the secretory and proliferative phase.

**DNAm networks are associated with menstrual cycle phase and endometriosis.** Weighted correlation network analysis (WGCNA)[33] was used to identify clusters (modules) of highly correlated DNAm sites that are associated with menstrual cycle phase or endometriosis case:control status. Thirty five DNAm modules were identified based on the WGCNA clustering of the DNAm profiles (Supplementary Fig. 9), eight of which were significantly associated (*p* < 0.05) with menstrual cycle phase and four with endometriosis case:control status (Fig. 2). Genes annotated to probes within clusters associated with cycle phase were enriched in 330 pathways including many related to cell migration, cell-cycle progression, cytoskeleton organization, transcription, and cell proliferation (Supplementary Data 8). These pathways included 70% of the pathways identified in the single site analysis. Genes annotated to probes within clusters associated with stage III/IV case:control status were

overrepresented in 74 KEGG pathways (Supplementary Data 9). The top pathways grouped into cellular proliferation, neuronal-related, ECM-cell interaction, and cancer pathways, including MAPK, Wnt, calcium and Hippo signaling, focal adhesion, axon guidance, and breast and gastric cancer (Supplementary Data 9; Fig. 2). The genomic locations for the module sites are provided in Supplementary Data 10.

**Genetic regulation of DNAm in endometrium.** To assess the effect of genetic variation on DNAm in endometrium, and identify endometrial mQTLs, we integrated DNAm data at the 759,345 sites with genotype data for 5,290,992 SNPs from 658 samples with European ancestry (Table 1; Supplementary Fig. 1). A total of 12,242,158 significant *cis*-mQTLs were identified in endometrium across 107,627 DNAm sites following Bonferroni correction (*p*-value < $1.7 \times 10^{-11}$). Stepwise regression

**Table 2 DNAm sites significantly associated with menstrual cycle phase.**

| Comparison | Direction | Number of DNAm sites ($p$-adj < 0.05) | DMRs (FDR < 0.1) |
|---|---|---|---|
| ESE_LSE | Down | 56 | 85 |
| | Up | 292 | |
| ESE_MSE | Down | 24 | 86 |
| | Up | 291 | |
| ESE_PE | Down | 1579 | 997 |
| | Up | 1821 | |
| LSE_MSE | Down | 1 | 0 |
| | Up | 4 | |
| LSE_PE | Down | 1878 | 897 |
| | Up | 1495 | |
| MSE_PE | Down | 4515 | 2371 |
| | Up | 3293 | |
| MSE_Menstrual | Down | 108 | 47 |
| | Up | 118 | |
| ESE_Menstrual | Down | 19 | 14 |
| | Up | 38 | |
| LSE_Menstrual | Down | 23 | 10 |
| | Up | 50 | |
| PE_Menstrual | Down | 23 | 5 |
| | Up | 42 | |
| PE_SE | Down | 5157 | 3005 |
| | Up | 4497 | |
| SE_Menstrual | Down | 69 | 30 |
| | Up | 94 | |

The number and direction of effect of DNA methylation (DNAm) sites significantly differentially methylated between menstrual cycle phases across all 984 samples regardless of endometriosis case:control status. Cycle phases include early secretory (ESE), mid secretory (MSE), late secretory (LSE), proliferative (PE), aggregated secretory phases (SE), and menstrual. The direction of effect between the cycle phases are annotated as *Down* if the methylation at the DNAm site is lower in the first phase and *Up* if the methylation is higher.

identified 118,185 independent *cis*-mQTL signals, evidence that some sites are regulated by multiple independent genetic variants. In total, mQTLs for 90,856 DNAm sites were not reported in endometrium previously[25]. There was evidence of mild genomic inflation in test statistics (lambda = 1.16). This is, however, consistent with inflation observed in differential DNAm models and is likely the result of residual variation from unknown mediators of variation in DNAm in endometrium. We were able to replicate 98% of endometrial mQTLs previously reported in a small preliminary study ($n = 66$)[25]. The proportion of variance captured by DNAm sites can be driven by genetics given the presence of endometrial mQTLs. To estimate of the proportion of variance captured by DNAm, independent of known genetic regulation, DNAm sites with mQTLs were removed and the variance captured by DNAm was re-calculated. In the absence of genetically regulated DNAm sites 15.4% of the variance was captured by DNAm (Table 2).

**Shared genetic regulation between endometrium and other tissues**. To investigate shared vs. tissue-specific genetic regulation, we compared *cis*-mQTLs found in endometrium with those reported in publicly available mQTL datasets including, blood[34,35], skeletal muscle[36], adipose tissue[37], and brain[38]. 78% of mQTLs identified in endometrium were also significant in blood and had the same direction of effect. In total, endometrial mQTLs for 3,480 DNAm sites were not observed in blood (Min et al.[35] = $p < 1 \times 10^{-8}$; Hannon et al.[34] = $p < 5 \times 10^{-8}$), brain ($p < 5 \times 10^{-8}$), muscle ($p < 5 \times 10^{-8}$), or adipose tissue (corrected $p$-value < 0.05) and may represent tissue-specific genetic regulation (Supplementary Data 11). An additional 19,317 DNAm sites

with significant mQTLs in endometrium on the EPIC array were not significant in blood. Genes annotated to the 3480 endometrium-specific DNAm sites were enriched in several Hallmark pathways including epithelial mesenchymal transition, androgen, and estrogen response, and KEGG pathways including ECM interactions and cell adhesion (Supplementary Data 12).

**Effect of genetic regulation is dependent on disease status and menstrual cycle phase**. To investigate if genetic effects on methylation differ between biological states in endometrium, we conducted context-specific mQTL analyses testing for differences in the independent endometrial cis-mQTL effects between women with and without endometriosis and between proliferative and secretory cycle phases. Context-specific analyses across all 118,185 mQTLs identified two with effects that were significantly ($p < 4.2 \times 10^{-7}$) associated with endometriosis status. Similarly, we found evidence of significant association between the effects of 12 mQTLs and menstrual cycle phase (Supplementary Fig. 10; Supplementary Data 13). mQTLs significant in the context-specific analysis suggests that the effect was larger and/or more variable in either cases or controls or proliferative or secretory samples. Larger sample sizes would be required to validate these context-specific effects.

**Genetic regulation of DNAm and transcription is associated with endometriosis**. To identify any associations between genetic regulation of DNAm in endometrium and risk of endometriosis, we integrated summary statistics from both endometrial mQTLs and an endometriosis genome-wide association (GWA) meta-analysis including 31,021 cases and 524,419 controls (Supplementary Data 20), using Summary data-based Mendelian Randomization (SMR)[39]. We identified 41 unique genetic variants that were associated with methylation at 45 DNAm sites and risk of endometriosis (Supplementary Data 14). Of the 45 mQTLs associated with endometriosis, 15 remained associated when cases were restricted to those with stage III/IV disease along with 6 mQTLs not associated with overall endometriosis (Supplementary Data 14). We were able to replicate published associations between methylation at DNAm sites near *GREB1* (Fig. 3) and *KDR* and endometriosis[25]. Out of the 51 mQTLs associated with endometriosis, 30 were also associated with the expression of 18 genes in various GTEx tissues, blood, and endometrium (Supplementary Data 14, 15). Approximately 50% of mQTLs associated with eQTLs had the opposite direction of effect whilst the remaining had the same direction consistent with previous observations and the hypothesis that the binding affinity of transcription factors and repressors on promoters and enhancers can be affected by DNAm[40]. Examples of DNAm sites affecting different genes in different tissue (eg. CDC42 in blood, WNT4 in thyroid and LINC00339 in endometrium) and different effects on the same gene in different tissues were also observed (e.g. ADK in lung vs colon). Expression of several of these genes (*LINC00339, CDC42, GDAP1, FGD6, SRP14*) has been associated with risk of endometriosis previously[10,24,41,42]. Expression of four of the genes (*EEFSEC, GDAP1, ADK,* and *SKAP1*) was also significantly associated with endometriosis risk when SMR analyses were conducted using the eQTL and GWAS summary statistics (Supplementary Data 16). Note it was not a requirement that the lead mQTL SNP and eQTL SNP associated with endometriosis be the same; however, they were in high linkage disequilibrium (LD) in European populations ($r^2 > 0.8$).

EpiMap and H3K27Ac HiChIP libraries were used to functionally annotate SNPs and DNAm sites significantly associated with endometriosis. Evidence of enhancers was found for 19 SNPs and 40 DNAm sites (Supplementary Data 14).

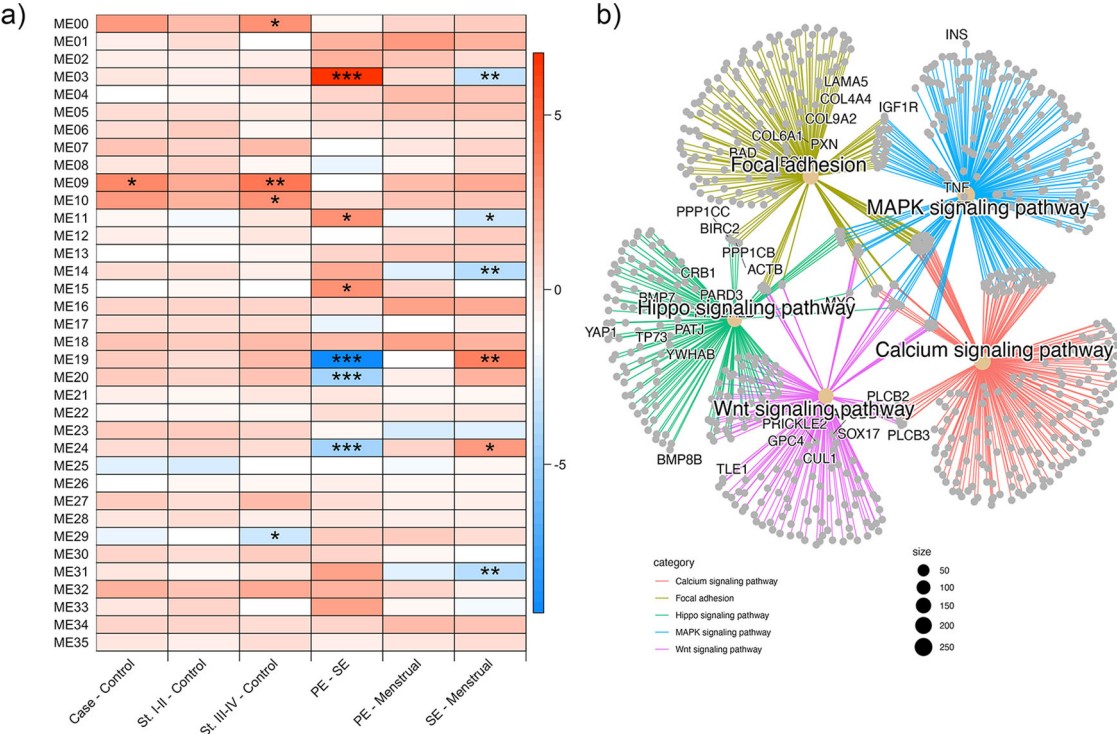

**Fig. 2 Network analysis. a** Modules of DNA methylation (DNAm) sites defined using WGCNA and their association with endometriosis status and menstrual cycle stage. * represents the degree of significance (***p-value = 0–0.001; **p-value = 0.001–0.01; *p-value = 0.01–0.05). **b** Network map of the top 5 most significantly enriched pathways for genes annotated to modules associated with endometriosis stage III/IV case:control. A total of 984 samples were included in these analyses (menstrual n = 50, proliferative (PE) n = 473, secretory (SE) n = 461, early secretory (ESE) n = 122, mid secretory (MSE) n = 209, late secretory (LSE) n = 108; cases n = 637 (stage I/II n = 344, stage III/IV n = 286), controls n = 347).

Similarly, evidence of promoters was found for 6 SNPs and 15 DNAm sites (Supplementary Data 14). Based on EpiMap predicted tissue-specific enhancer-gene links and location of predicted promoters, we identified 27 target genes with links to the enhancers and promoters (Supplementary Data 14). Target genes identified include those previously implicated in endometriosis risk on chr1p36.12. mQTLs in this region were in predicted enhancer regions and promoter regions of *WNT4* in uterus and were associated with expression of nearby *CDC42* and *LINC00339* (Supplementary Fig. 11). Other previously unreported examples include *EEFSEC* on chr3q21.3 and a cluster of HOXC genes on chr12q13.13 (Supplementary Fig. 12). mQTLs in the chr3q21.3 region associated with expression of *EEFSEC* were located in a predicted enhancer region in uterus, and one SNP (rs2999046) was located in an active chromatin area predicted to interact with *EEFSEC* and *RUVBL1* in immortalized endometrial cells (Fig. 4; Supplementary Data 14).

In total, 45 genes were annotated to endometriosis-associated mQTLs using epigenetic data and eQTLs, 16 of these genes are in loci not previously associated with both expression/methylation and endometriosis risk (Supplementary Data 14). These 45 genes had similar expression across reproductive tissues (ovary, fallopian tube, uterus, cervix), shown by gene expression clustering in FUMA (Supplementary Fig. 13). Genes were enriched for androgen receptor targets and in several GO adhesion-related pathways (Supplementary Data 17) and multiple protein-protein interactions were identified in STRING, the network having significantly more interactions than expected ($p = 7.71 \times 10^{-3}$) (Supplementary Fig. 14).

Variants significantly associated with both DNAm and risk of endometriosis are also associated with other traits and diseases providing insight into possible shared underlying pathways. Other reproductive traits associated with these SNPs include

uterine fibroids, female genital prolapse, ovarian cancer, age at menarche and bilateral oophorectomy (Supplementary Data 14).

**Differential DNAm by endometriosis-related surgical and pain sub-phenotypes.** We investigated differential DNAm signatures associated with endometriosis subphenotypes: disease stage (I/II and III/IV), lesion type (endometrioma, superficial, deep disease) and three different pelvic pain symptoms: dyspareunia (pain with intercourse), acyclic pain, and dyschezia (bowel movement pain). To minimize the number of tests conducted and maximize the a-priori chance of detecting signals, we focused the analysis on 11,698 sites within 500Kb± of 44 SNPs previously found to be genome-wide significantly associated with endometriosis[10] (Supplementary Data 18).

Adopting a genome-wide Bonferroni-corrected threshold ($p < 4.3 \times 10^{-6}$) across all comparisons, only one DNAm site—located in *ADAMTSL2* on chromosome 9 (cg13469396)—was significantly different, comparing cases with acyclical pain only vs. NUPP controls. When a less stringent threshold was applied ($p < 0.05$/N of DNAm sites per GWAS locus) a total of 66/11,698 DNAm sites showed differential DNAm with at least one of the sub-phenotypes (Fig. 5, Supplementary Data 19). These included five differentially methylated sites in stage III/IV endometriosis vs. controls, one in stage I/II endometriosis vs. controls (Fig. 5: blue panel, Supplementary Data 19). No sites reached statistical significance in stage III/IV vs. stage I/II. When considering more detailed surgical features of endometriosis (e.g., endometrioma, superficial, deep), 29 sites were differentially methylated (Fig. 5: purple, green and red panels). Fourteen of these had mQTLs; however, no mQTL-regulating SNPs were in high LD ($r^2 > 0.8$) with a lead endometriosis-associated risk variants in these regions.

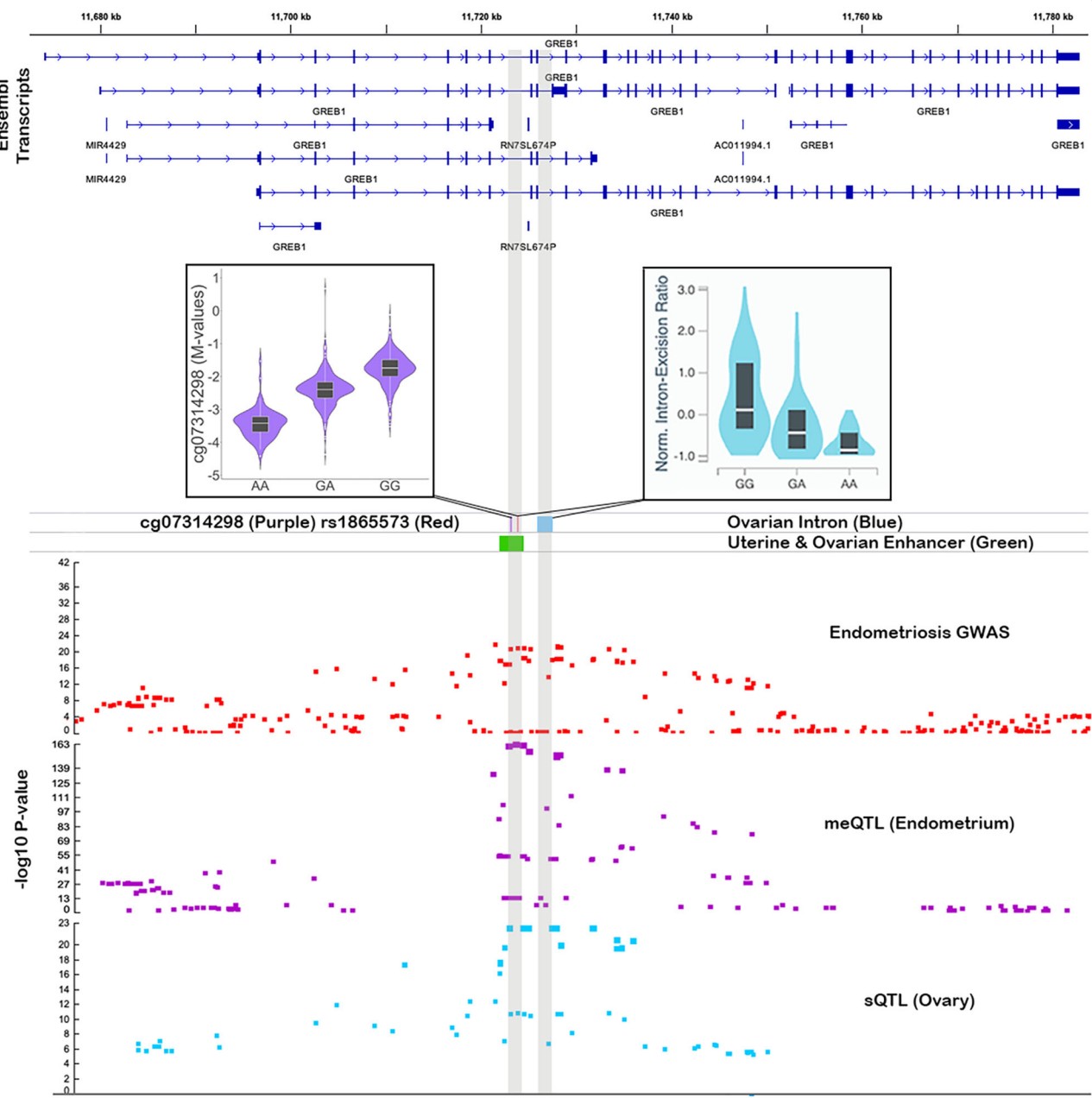

**Fig. 3 GREB1 mQTL associated with endometriosis risk.** The top panel shows ensemble transcripts present in the locus. The bottom panel consists of association plots, each point is a SNP plotted according to its genomic position on the x-axis and −log10 p-value for its association with endometriosis (red) DNA methylation (DNAm) at cg07314298 (purple) and GREB1 splicing in ovary (blue) on the y-axis. The position of the associated spliced intron (blue) is featured in the middle panel alongside the position of the significant SMR mQTL SNP (red) and DNAm site (purple) and the position of a predicted enhancer in uterus and ovary (green). Boxplots show the difference in DNAm and intron excision according to the genotype at rs1865573. A total of 658 European samples were used in the mQTL analysis to test for associations between genotype and DNAm in endometrium.

When considering pain subphenotypes, 23 DNAm sites were associated with dyspareunia and acyclical pain, including 18 with mQTLs (Fig. 5: dark blue panel and pink panel; Supplementary Data 19). One DNAm site associated with acyclical pain was located in a predicted enhancer in uterus with links to WT1, a transcription factor that plays an important role in gonad development and cellular development and survival and has been linked to endometriosis[43,44] (Fig. 5: pink panel, Supplementary Data 19).

Lastly, we focused on cases with endometriosis-associated painful bowel movement (dyschezia), as a symptom previously shown to be predictive of endometriosis[45] and hypothesized as a potential indicator of rectovaginal deep disease. Analysis of all dyschezia vs. controls revealed the largest number of differentially methylated sites—11 in total (significantly more than expected by chance: binomial test p-value = $1.41 \times 10^{-5}$) (Fig. 5: yellow panel, Supplementary Data 19). One of the differentially methylated sites, cg01340163, located in the KDR gene promoter on 4q12, has an associated mQTL in endometrium where the mQTL SNP (rs12331597) is in high LD ($r^2 = 0.97$) with an endometriosis-associated variant in this locus (rs1903068) (Supplementary Fig. 15).

There was no evidence for systematic differences in effect sizes across all 66 DNAm sites comparing endometrioma or deep

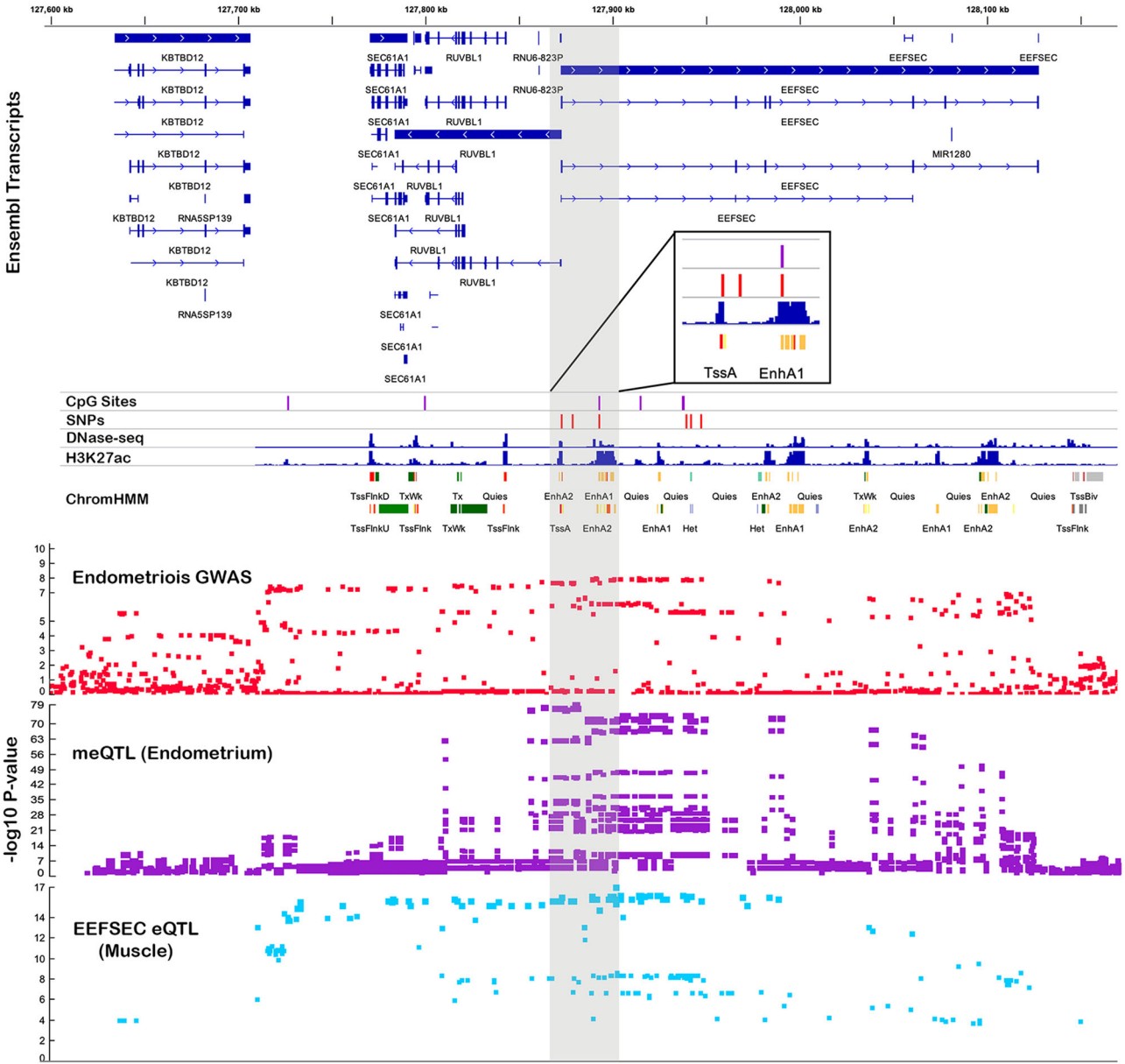

**Fig. 4 *EEFSEC* mQTL associated with endometriosis risk.** The top panel shows ensemble transcripts present in the locus. The bottom panel consists of association plots, each point is a SNP plotted according to its genomic position on the *x*-axis and −log10 *p*-value for its association with endometriosis (red) DNA methylation (DNAm) at six SMR significant DNAm sites (purple) and *EEFSEC* expression in muscle (blue) on the *y*-axis. The position of the significant SMR mQTL SNPs (red) and DNAm sites (purple) is featured in the middle panel above DNase-seq peaks, H3K27ac peaks and predicted chromatin marks in uterus. A total of 658 European samples were used in the mQTL analysis to test for associations between genotype and DNAm in endometrium.

endometriosis with stage I/II vs. stage III/IV (Supplementary Fig. 16a, b); or comparing effect sizes for dyschezia, dyspareunia, or acyclic pain with any of the surgical subtypes (Supplementary Fig. 17a–c).

## Discussion

The objectives of this study were to analyze epigenetic signals in the endometrium associated with endometriosis and genetic regulation of epigenetic signals in genomic risk regions for this disease. Epigenome-wide association studies (EWAS) test associations between differentially methylated DNAm sites and complex diseases[46,47]. The tissue under analysis is crucial to detect biologically relevant epigenetic changes and we conducted this study in endometrium as a strong candidate tissue for the origin of cells that

initiate endometriosis lesions[48,49]. Our single site analysis did not reveal any statistically significant differences in DNAm in endometrium from women with or without endometriosis. Differences in methylation at previously identified candidate genes, including *HOX-A10*[19], *PR*, *ESR1*[13,20], *CYP19*[14], *SF-1*[15,16], *COX-2*[17] and *DNMTs*[18], were not replicated in this study. Failure to replicate these candidates likely reflects differences between genome-wide and candidate gene approaches, use of eutopic versus ectopic tissue and small sample sizes used previously.

Variation in DNAm between individuals and populations can arise as consequences of environmental exposures, stochastic and genetic perturbations[50,51] and both causes and consequences of disease[52–54]. Our study of methylation signals in endometrium has a ten-fold increase in cases compared with previous studies[25] and

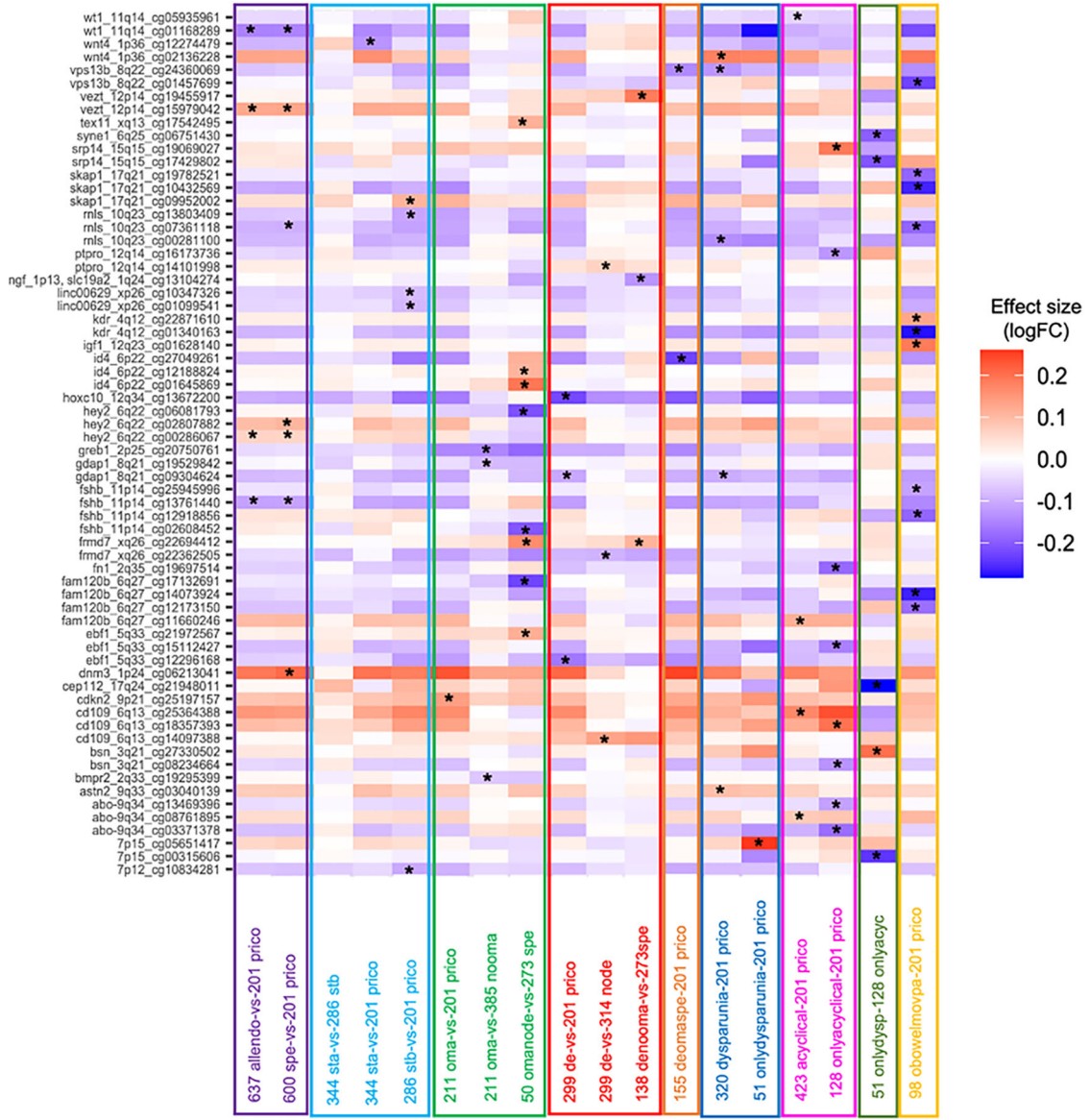

**Fig. 5 Heatmap for effect sizes of 66 differentially methylated sites across 17 sub-phenotype comparisons.** DNA methylation (DNAm) sites are presented as rows and differential DNAm analysis for each phenotype is presented as columns. * Denotes statistically significant DNAm sites passing the locus specific Bonferroni-based multiple-testing correction ($p < 0.05$/N of DNAm sites per GWAS locus). allendo: all endometriosis cases ($n = 637$), prico: NUPP controls ($n = 201$), spe: superficial lesions ($n = 600$), sta: rASRM stage I/II disease ($n = 344$), stb: rASRM stage III/IV disease ($n = 286$), oma: endometrioma ($n = 211$), omanode: cases with endometriomas but no deep lesions ($n = 50$), de: deep lesions ($n = 299$), denooma: cases with deep lesions but no endometriomas ($n = 138$), deomaspe: cases with co-occurrence of superficial lesions, endometriomas and deep lesions ($n = 155$), dyspareunia: cases with dyspareunia ($n = 320$), onlydyspurunia: cases with dyspareunia but no acyclic or dychezia ($n = 51$), acyclic: cases with acyclic pelvic pain ($n = 423$), onlyacyclical: cases with acyclic pelvic pain but no dyspareunia or dyschezia ($n = 128$), obowelmovpa: cases with only dyschezia ($n = 98$).

suggests individual changes in methylation signals associated with endometriosis will be small. For example, a recent comparison of 924 monozygotic and 1033 dizygotic twin individuals identified a signature of 243 epigenome-wide significant differentially methylated positions between MZ and DZ twins[55]. Effects of nominally significant differentially methylated sites between endometriosis cases and controls in our study (max delta-beta = 0.06, mean = 0.01) were much smaller than absolute differences in the MZ twin signature which ranged from 0.3 to 6%, with a mean of 2.2%[55]. This suggests even larger sample sizes would be needed to achieve sufficient power to detect the smaller effects.

Previous studies have estimated the amount of variation in complex traits captured by methylation, with estimates ranging from 0% for height, 6.5% for BMI[56] and 15–31% for amyotrophic

lateral sclerosis[57], to almost 100% for age. For our study, we have done our best to match our case and control groups by age and therefore have chosen not to include age as an additional surrogate variable. We do believe that the turnover of the uterus likely precludes age as a factor in DNAme, but this is something that could be explored in future studies. Although we did not identify significant individual signals, DNAm in endometrium was estimated to capture 15.4% of the variance in endometriosis case control status. On the flip side, the methylation differences attributable to the disease that we observe are quite low (0.03–0.08%). This suggests many small effects of methylation across the genome, or within pathways or networks of genes, may contribute to this variation. The ability to detect differentially methylated sites associated with disease may be attributed to

amount, duration, and/or time of critical exposures and the expected effect size of these exposures. The critical window to quantify etiologically relevant differential methylation at DNAm sites in endometrium associated with risk may be prior to overt symptoms or diagnosis. Moreover, endometriosis case vs. control differences may be affected by cell-type composition of tissues, endometriosis disease heterogeneity or mediated by previous use of medications or behavioral changes to reduce symptoms[58,59]. Single-cell RNA-sequencing and deconvolution of bulk endometrial samples has highlighted signatures for cellular subtypes in endometrium that likely contribute to variation between samples[60–62]. However, in the absence of reference datasets for these cell-types appropriate corrections for cell-type composition cannot be applied to methylation analyses. Future investigations into cell type-specific effects may be facilitated by emerging technologies in single-cell epigenetics.

In support of the effects of disease heterogeneity, we found evidence of genome-wide differences in methylation at four DNAm sites when cases were restricted to stage III/IV disease despite reduced power. Larger molecular effects in severe disease are consistent with genetic studies that show evidence of larger genetic effects in stage III/IV endometriosis[10,32] and transcriptomic studies showing differences in endometrium of patients with stage III/IV versus stage I/II endometriosis support lower implantation and pregnancy rates in in those with more advanced disease in both natural and assisted reproduction cycles[63]. One of the genes annotated to these differentially methylated sites, *EEF2* (eukaryotic translation elongation factor 2), is downregulated in ectopic endometriosis lesions[64]. Similarly, *TNPO2* (Transportin 2), a nuclear transport receptor highly expressed in the ovary, uterus, testis and brain, is a predicted target of a microRNA found to be differentially expressed between ectopic and eutopic endometrial tissue[65], and overexpressed in human ectopic endometrial epithelial cells[66]. *ELAVL4* (ELAV Like RNA Binding Protein 4) encodes an RNA-binding protein with roles in post-transcriptional regulation of mRNAs and is predominantly expressed in neurons, pituitary, and testis and has been associated with neurodegenerative diseases, type 2 diabetes and prognosis of endometrial adenocarcinoma[67,68]. Interestingly, network analysis identified modules of DNAm sites associated with endometriosis case:-control status. Genes annotated to these modules were enriched in WNT and MAPK signaling, adhesion and cancer pathways. Both the WNT and MAPK signaling pathways have been associated with endometriosis in previous expression studies and have been identified as potential treatment targets[69–72]. Consequently, larger sample sizes will be needed to identify changes in methylation signals in disease subtypes and account for differences across the menstrual cycle.

Numerous DNAm sites distributed across the genome are differentially methylated in human endometrium across the menstrual cycle, with variation likely driven by cyclicity of cell differentiation and turnover, and tissue-specific response to circulating steroid hormones[8,73]. Differences observed in this study are consistent with those reported in smaller studies on endometrium with the addition of many more differentially methylated sites as a result of increased power[21,25,73–76]. We validate previous observations of changes in methylation of DNA methyltransferases (e.g. *DNMT3a, EZH2*), hormone response genes (e.g. *HOXA9, MYO3A*) and genes associated with transcription regulation (e.g. *RUNX3*), decidualization (e.g. *WNT4, ZBTB16, GREB1, HAND2, STAT5A, HOXA10*) and embryo implantation (e.g. *MAPK14, ZMIZ1, PLXNA4*)[73]. A recent study also highlighted the importance of epigenetic modifications during endometrial decidualization and whilst they did not observe methylation decidualization-associated genes (*WNT4,*

*HAND2, STAT5A*) they did report increased level of histone H4 hyperacetylation in these regions during in vitro ESC decidualization[77]. Evidence suggests that changes in DNAm profiles across the menstrual cycle is a phenomenon that may be specific to endometrium and is not observed in blood[25,73].

Pathways enriched for genes annotated to DNAm sites differentially methylated between the proliferative and secretory phases reflect changes in endometrial function including cell cycle progression (RHO and CDC42 GTPase cycle) and tissue remodeling (regulation of actin cytoskeleton, ECM organization). Seventy-eight genes annotated to DNAm sites differentially methylated across the menstrual cycle are present on the Endometrial Receptivity Array (ERA)[78], which contains 238 genes in total, representing a significant enrichment ($p = 1.46 \times 10^{-6}$). One of these genes (*ABCC3*) was annotated to a DNAm site differentially methylated between early and mid-secretory phases with lower methylation in the mid-secretory phase consistent with previous reports of higher gene expression[78], suggesting that changes in DNAm may also influence endometrial receptivity to embryo implantation and/or the accuracy of estimating the window of implantation using these genes. Indeed, a recent review highlighted changes in methylation observed during the implantation window and the association of alterations in methylation with defective receptivity and implantation failure[73,79]. Interestingly, prior to correction with SVs we also observe variation in endometrial methylation associated with genetic ancestry however, given the small sample size of the non-European ancestral groups we lacked power to test for differences between ancestries. Future work in more diverse populations is instrumental in investigating ancestry-dependent effects. Consideration of changes in DNAm across the menstrual cycle is vital when assessing effects of regulation of molecular mechanisms on endometrial pathologies and fertility traits.

Epigenetic regulation in endometrium has the potential to influence vital aspects of endometrial function, female fertility, and disease. Integration with genetic data identified 118,185 independent *cis*-mQTLs including >3000 potential tissue-specific mQTLs and numerous mQTLs associated with endometriosis risk. We identified 51 mQTLs, distributed in 21 genomic loci, that were significantly associated with endometriosis risk from the SMR analysis. Associations between endometriosis and QTLs at 10 loci have not previously been reported to the best of our knowledge[10,24,25,41]. These results provide evidence that endometriosis-associated variants have functional effects on endometrial methylation for ~40% of endometriosis risk loci identified in the most recent endometriosis GWAS[10]. This is more than that observed previously for eQTLs[24,41], suggesting the genetic contribution to endometriosis may have a bigger effect on methylation and/or larger eQTL studies are needed in endometrial cell types to detect significant effects at these loci.

Examples of new evidence for known endometriosis target genes that warrant further discussion include *GREB1*, and genes on chromosomes 1, 6, 12, and 17. *GREB1* is an estrogen- and androgen-regulated gene which mediates cell proliferation and migration, and induces EMT in ovarian cancer[80] and hormone-stimulated proliferation in breast cancer[81]. mQTLs significantly associated with endometriosis featured DNAm sites and SNPs located in predicted enhancers that interact with *GREB1* in uterus and ovary including rs1865573 that is associated with a splicing (s)QTL for *GREB1* in ovary reported in GTEx[82], suggesting a possible role of alternative splicing of *GREB1* in endometriosis susceptibility. Signals for mQTLs significantly associated with endometriosis on chromosome 1p36.12, including two potentially endometrial-specific mQTLs, were located in regulatory regions and associated with the expression of known genes of interest *LINC00339, CDC42,* and *WNT4*[10,24,41,42]. Similarly, potential

endometrial-specific endometriosis-associated mQTLs lie in a predicted promoter for *CCDC170* and enhancer regions that may mediate regulatory effects on multiple genes in the chromosome 6q25.1–6q25.2 risk region. Expression of genes in this region is known to be highly correlated with nearby *ESR1* expression in endometrium[10,83]. Finally, endometriosis-associated mQTLs associated with expression of *GDAP1, SRP14,* and *HOXB9* are in moderate LD ($r^2 > 0.6$) with SNPs significantly associated with endometriosis and with eQTLs for these genes in the recent endometriosis meta-analysis[10].

Genetic regulation of DNAm in endometrium may also influence other reproductive pathologies including cancer. Variants significantly associated with both endometriosis risk and DNAm in endometrium on chromosomes 10p12 and 12q22 were in LD with rs7084454 ($r^2 = 0.89$) and rs6538618 ($r^2 = 0.99$) respectively, risk SNPs shared between endometriosis and epithelial ovarian cancer. These mQTLs are located in predicted promoter regions for *MLLT10* and a bidirectional promoter for *VEZT* and *FGD6* and the latter is associated with expression of both genes in endometrium[84]. Epidemiological studies report an increased risk of ovarian cancer in women with endometriosis and recent genetic studies report a strong genetic correlation and causal relationship between endometriosis and specific epithelial ovarian cancer subtypes[84–86].

Characterizing endometrial specific genetic regulation is important when investigating functional effects of gynecological disorders and fertility traits. The majority of mQTLs we identified in endometrium have also been reported in blood, suggesting the genetic regulation of these DNAm sites is shared between tissues, consistent with tissue overlap reported in previous studies[25,38]. However, we did identify genetic regulation for 3480 DNAm sites that have not been previously reported in other tissues. Genes annotated to these potential endometrium-specific mQTLs were enriched in pathways known to be important in endometrial biology including estrogen and androgen response, regulation of epithelial to mesenchymal transition (EMT), cell adhesion, calcium signaling, and ECM receptor interaction, important for endometrial tissue integrity and differentiation, decidualization, embryo implantation, and establishment of early pregnancy[87–91]. To our knowledge this study is the largest collection of DNAm data from human endometrium with associated detailed clinical information, a powerful resource for advancing reproductive medicine.

An atlas of genetic effects on DNAm in blood generated from 32,851 individuals demonstrated that although many genetic signals are shared between DNAm sites and complex traits, altered DNAm signals are not necessarily on a causal path from genotype to phenotype[35]. The complex relationship may instead be driven by genetic effects on other regulatory factors that affect DNAm and the trait through diverging pathways (horizontal pleiotropy), reverse causation, cell-type or context specific effects or a non-linear causal path involving several interactions of regulatory features[35]. Context-specific genetic effects on DNAm in endometrium for several mQTLs varied between menstrual cycle stages and endometriosis cases and controls, warranting replication in independent datasets. Although we are unable to distinguish causal and pleiotropic pathways, we identified several endometrial mQTLs that were statistically significantly associated with endometriosis risk, providing further functional evidence for endometriosis target genes than previous studies in eQTLs[10,24,41].

New endometriosis target genes detected include *EEFSEC* (eukaryotic elongation factor, selenocysteine-TRNA specific) on chromosome 3q21.3, *SRD5A3* (steroid 5 α-reductase 3) on chromosome 4q12, *ADK* (adenosine kinase) on chromosome 10q22.2, and HOXC cluster on chromosome 12. Endometriosis-associated mQTL SNPs on chromosome 3q21.3 in *EEFSEC* are in

LD with variants previously associated with gestational length, spontaneous preterm birth[92] and age at menarche[93,94], risk factors for endometriosis patients. Epigenetic marks and chromatin interactions suggest these mQTLs fall in *EEFSEC* promotor and enhancer regions in uterus and ovary, where the gene is highly expressed[95], and immortalized endometrial cells[96,97]. In addition, methylation at DNAm sites located near the *KDR* promoter on chromosome 4 are associated with endometriosis risk and expression of nearby *SRD5A3* in blood. *KDR* encodes vascular endothelial growth factor receptor 2 (*VEGFR-2*), a major mediator of angiogenesis, proliferation, migration, and differentiation of endothelial cells. Dysregulation of angiogenic activity in endometriotic lesions and the eutopic endometrium of women with endometriosis and is mediated by VEGF signaling[98,99].

Several mQTLs specifically associated with severe (rASRM stage III/IV) endometriosis including five on chromosome 10q22.2 are also associated with eQTLs for *ADK*, an enzyme that regulates concentrations of extracellular adenosine and intracellular adenine nucleotides. Evidence from epigenetic marks suggests that one of these DNAm sites is located in the promoter region of *ADK* in fibroblasts, uterus and ovary and is in an anchor point for chromatin interaction with *ADK* in immortalized endometrial cells. Notably, adenosine nucleotides as targets for endometriosis pain management are currently under investigation[100]. Whether endometrial mQTLs or eQTLs for *ADK* influence eutopic endometrial tissue-specific inflammation in women with endometriosis is an attractive hypothesis and warrants further study[100].

Detailed differential DNAm sub-phenotype analyses, based on surgically visualized disease phenotypes, endometriosis-associated pain symptomatology, and rASRM staging, could facilitate understanding mechanisms underlying particular biological subtypes of endometriosis and lead to stratified diagnosis and therapeutic target discovery. To maximize power of detection, given the relatively small sample size of sub-phenotype groups ranging from 50–600 cases, we focused sub-phenotype analyses on differential DNAm signatures linked to 44 endometriosis-associated GWAS loci[10]. The results showed multiple differentially methylated sites associated with surgical phenotypes in particular with rASRM stage III/IV and endometrioma, suggesting different genetic regulatory mechanism for this particular sub-type of endometriosis. Several of the associated DNAm sites were located in enhancer and promoter regions in uterus including enhancers predicted to interact with *WT1* on 11q14, important in gonad development, and promoters for genes associated with genitourinary anomalies (*MMPED2*). Immunostaining evidence suggests *WT1* is downregulated in the eutopic endometrium from women with endometriosis but is expressed in neurons of deep endometriosis[43,44]. Moreover, we identified that regulation of DNAm at cg24360069, in *STK3* on 8q22, was associated with a greater extent of endometriotic pelvic disease, presence of dyspareunia, and had an mQTL in high LD with the lead GWAS variant at this locus. *STK3* encodes Serine/threonine-protein kinase 3, which is part of the Hippo pathway that plays an important role in tumor suppression by reducing cellular growth and promoting apoptosis. Notably, for the pain subphenotypes, dyschezia had the largest number of DNAm sites. One of these, located in the promoter of *KDR*, had an mQTL in LD with the lead GWAS variant. The single genome wide significant DNAm site, associated with acyclical pain, was located within an exon of *ADAMTSL2* on chromosome 9, a disintegrin and metalloproteinase with thrombospondin motifs and is suspected to interact with latent transforming growth factor beta binding protein 1.

This study presents the largest and most comprehensive analysis of regulation of DNAm in the human endometrium. Menstrual cycle stage is a major source of variation in DNAm in

endometrium—providing strong support for hormonally driven changes in DNAm across the cycle being associated with known changes in gene expression and pathways responsible for endometrial physiology, function, and dysfunction. We found no evidence for large site-specific DNAm differences between endometriosis cases and controls. However, variation in DNAm signals captured 15.4% of the variance in endometriosis case control status and differences at four DNAm sites were significant when cases were restricted to stage III/IV disease. We also report previously unreported disease-associated genetic effects on DNAm in endometrium that provide new insights into epigenetic regulation of pathways important for endometrial biology and function and identify target genes with a potential role in the causal pathway between genetic variation and endometriosis pathogenesis. Functional validation of these genomic targets may yield new therapeutic targets with the potential to disrupt pathogenic processes. Possible differences in DNAm associated with disease sub-phenotypes highlights the need to generate larger datasets with sample collection timed with menstrual cycle phase and comprehensive clinical information. Validation of methylation profiles associated with sub-phenotypes of disease may also allow molecular subtyping of disease that can be associated with patient outcomes to guide personalized disease management. Findings from this study will direct future endometriosis research and datasets generated will be a valuable resource for subsequent investigations into tissue-specific effects of methylation on endometrial biology and disease, and development of potential novel, targeted therapeutics.

## Methods

**Ethics approval and consent to participate.** All participants provided the site-specific study investigator with informed consent. All patient data were de-identified and followed HIPAA and the Convention of the Declaration of Helsinki. This study was approved by the institutional review boards of UCSF (Administrative Multi-Principal Investigator site), Michigan State University (Multi-Principal Investigator site), University of Oxford, University of Melbourne and University of Edinburgh.

### Sample collection and clinical data standardization

*Study participants.* Endometrial tissue from 679 surgically diagnosed endometriosis patients (cases), 389 controls without endometriosis and six participants with unconfirmed endometriosis status were recruited through the University of California San Francisco, California (UCSF, $n = 480$ samples), University of Melbourne, Melbourne, Australia (UM, $n = 315$ samples), Endometriosis CaRe Centre in Oxford, Oxford, UK (ENDOX, $n = 193$ samples), and EXPPECT Centre, The University of Edinburgh (EDIN), Edinburgh, Scotland, UK ($n = 86$ samples), with collection at all sites using the World Endometriosis Research Foundation Endometriosis Phenome and Biobanking Harmonization Project (WERF EPHect) standardized protocols for tissue collection and processing, and participant characteristics and clinical annotation (Supplementary Data 1)[26–28]. Participants were restricted to those who had not been on contraceptive steroids or gonadotropin releasing hormone analogues for 3 months or more prior to endometrial sampling, had regular cycles (defined as 24–35 days in length) and no evidence of endometrial hyperplasia or cancer. Women without visualized endometriosis at the time of surgery or without a history of endometriosis were defined as controls: some of these had pelvic pain and documented uterine fibroids or other non-malignant gynecologic conditions including endometrial polyps, ovarian cysts, cervical abnormalities, and pelvic organ prolapse; a subset was defined as *no uterine or pelvic pathology* (NUPP) controls ($n = 201$) if they had no visualized or documented uterine or pelvic pathology. Controls were sourced from all four recruitment sites in roughly equal proportions to avoid bias (Supplementary Data 1). After quality control described in detail below, the study included DNA from a total of 984 endometrial tissue samples.

*Tissue collection and processing.* Each participant contributed a single endometrial tissue sample. Primary tissue samples were stored as fresh frozen (FF) specimens in liquid nitrogen or at −80 °C. Samples from the contributing sites were combined at UCSF into two batches, prepared consistently and randomized among plates. Batch I consisted of 767 samples (474 cases; 2 unknown; 291 controls) from UCSF, UM, and ENDOX. Batch II consisted of samples from additional recruitment and included 307 samples (205 cases; 4 unknown; 98 controls) from all four sites (Supplementary Fig. 1). The total number of cases and controls (all types) was 679 and 389 respectively. Samples in both batches were analyzed for genotyping and

DNAm using the same platforms. All DNA samples were adjusted to the same volume and concentration for sodium bisulfite conversion followed by DNAm analysis on the Illumina Infinium MethylationEPIC Beadchip (Illumina, San Diego, CA) at the University of Southern California Epigenome Center Core Facilities, Los Angeles, CA. They were assessed for amount and completeness of sodium bisulfite conversion using a panel of MethyLight reactions[9]. In brief, a MethyLight reaction for a genome-wide distributed multicopy *ALU* sequence that is bisulfite-dependent but DNAm-independent[101] assessed the integrity and quantity of the DNA samples. The completeness of bisulfite conversion was assessed by three bisulfite-dependent reactions measuring 0%, 50%, and 100% bisulfite conversion for each sample[101]. Samples were required to pass all quality control (QC) metrics [high DNA integrity (*ALU* CT < 25), 100% bisulfite conversion with no amplifications at 0% or 50% conversion] before quantitative assessment of DNAm on the Illumina Infinium platform.

Cycle phase was assigned using the criteria of Noyes et al.:[102] menstrual, early proliferative (EP), mid-proliferative (MP), late proliferative (LP), early secretory (ESE), mid-secretory (MSE), and late secretory (LSE) for all specimens. Not all cohorts had assigned proliferative substages and as such all proliferative samples were consolidated as *PE*. A small number of samples ($n = 23$) were not assigned a secretory substage and were assigned *SE*. If two phases were found in reports or on review of histology, the later phase was selected (e.g. LSE/Menstrual → Menstrual; PE/ESE → ESE, Interval → ESE). Serum estradiol (E₂) and progesterone (P₄) facilitated phase assignments, and sometimes two or more pathologists re-reviewed the histology. Unsuitable samples (inactive, atrophic, PE/SE, progestin effect, dyssynchronous) were excluded. *Benign* histology descriptor and *unknown* were assigned as *unknown* in the absence of last menstrual period (LMP) and/or serum E₂ and P₄ levels (assayed at the University of Virginia NIH Eunice Kennedy Shriver National Institute of Child Health and Human Development (NICHD) Ligand Core). Endometrial samples from women with a history of endometriosis but no disease identified at surgery were not considered as controls and were not suitable for DNAm quantification, and thus were excluded from that analysis. All four sites completed histology assessments and/or cycle phase determinations for their respective sample specimens.

*Participant characterization.* Variables across all site-specific datasets were combined and harmonized (listed in Table 1). The following woman-level covariates were included in statistical modeling. *Site*: A categorical variable with a unique value for each of the five sample contributing sites/institutions (UCSF, ENDOX, UM, EDIN). *Cycle phase*: A categorical variable with a unique value for each of the six menstrual cycle phases defined previously (Menstrual, PE, SE, ESE, MSE, LSE). *Endometriosis case:control status*: A binary variable assigning samples as either a case or control. *Sample plate*: A categorical variable with a unique value for each of the 12 sample plates used during processing. *Batch*: A binary variable assigning samples as either Batch I or II.

Endometriosis sub-phenotyping characteristics included: *rASRM endometriosis disease stage*: visualized at surgery most proximal to endometrial biopsy collection and defined by the rASRM endometriosis scoring system[29]. Stage data were used to create variables with three structures—continuous rASRM score, ordinal stages I, II, III, IV, and dichotomized as stage I + II and stage III + IV. For case patients for whom surgical documentation was noted as the rASRM stage category only, the variables were categorized as documented. However, four patients were defined in their surgical record as having visualized stage II-III and were assigned to the I–II dichotomized rASRM category. *Lesion type*: categorized according to the presence of at least one superficial peritoneal lesion, endometrioma, or deep lesion. Lesion types were binary variables coded as *any* peritoneal lesion, endometrioma, or deep lesion, regardless of co-occurrence of another lesion type and not mutually exclusive variables. *Pain*: binary variables for the presence or absence of dyspareunia, acyclic pelvic pain and dyschezia (see Supplementary Data 16 for detailed descriptions).

### Data quality control and processing

*DNAm data quality control.* DNAm data were generated using the Illumina Infinium MethylationEPIC Beadchip (Illumina, San Diego, CA) at USC Epigenome Center Core Facilities, with data on 865,859 DNAm sites for 767 samples in batch I and 307 in batch II (total $n = 1,074$). DNA from only one endometrial tissue sample per participant was included. DNAm QC and processing were conducted on the dataset through a series of steps (Supplementary Fig. 1). The *openSesame* function from the *SeSAMe*[103] R package, a pipeline for Illumina Infinium human methylation processing, was used to perform background correction with dye-bias and signal intensity normalization starting from the idat files, and Beta value generation. The following filtering steps for samples and DNAm sites/probes were carried out. Samples with a low overall intensity signal, defined as a median unmethylated or methylated signal <9, were removed from the dataset. In addition to this, samples were also filtered out if they had a detection $p$-value > 0.05 in more than 1% of DNAm sites and probe level quality control and processing were done by filtering out probes with a detection $p$-value > 0.05 in more than 10% of samples ($n = 1727$ probes in batch 1, $n = 1,461$ probes in batch II). Probes were masked based on the *SeSAMe* package. These included the 59 Illumina tagged probes and three classes of probes that could be potentially problematic or ambiguously mapped ($n = 104,671$ probes in batch I and $n = 104,752$ probes in batch II) as well

as chromosome Y probes ($n = 26$ probes). Finally, the *minfi*[104] function 'getSex' was used to generate a predicted sex for each sample, in order to ensure all samples were confirmed as genetically female based on the median values of measurements on the X and Y chromosomes ($n = 0$ samples were removed). The next QC stage involved probe removal of non-overlapping probes in the data between batch I and II ($n = 275$ probes). These sample and probe level QC steps resulted in a processed dataset of 759,345 DNAm sites for $n = 707$ and $n = 277$ endometrial samples in batch I and II, respectively, with a final total of 984 samples (from 637 cases and 347 controls) (Table 1).

*Genotyping data quality control.* Batch I and II samples that passed the DNAm QC were genotyped using the Axiom Precision Medicine Research array. Quality control of the data was conducted in three stages; (1) batch I samples ($n = 707$ individuals), (2) batch II samples ($n = 277$ individuals), (3) Merged batch I and II samples. In each stage (i) per-individual QC included identification and filtering of samples with genotype call rate < 95%, heterozygosity rate > 3 standard deviations away from the mean heterozygosity rate and removal of related (IBD > 0.200) samples, (ii) per-variant QC included filtering variants with MAF < 1%, call-rate >95% and HWE *p*-value < $1 \times 10^{-5}$. Post-QC combined datasets included 953 individuals (614 cases, 339 controls) and 621,613 variants. Post-QC, data from all ancestries was pre-phased using SHAPEIT2 and imputed using the 1000 Genomes reference (1000G P3v5) all together. Not all samples that passed the methylation QC also passed the genotype QC and as such only samples passing both were included in the subsequent mQTL analysis.

*Genetic ancestry determination.* Genetic ancestry of each study participant was determined using 1000 Genomes P3v5 reference data. In brief, principal component analysis was conducted with common markers between the study samples and the 1000G P3v5 reference data. Then, the first 10 principal components were plotted against each other to identify population clusters in our data against the 1000G P3v5 reference data with five super populations including European, Eastern Asian, American, African, Southern Asian. Any samples that did not cluster with the 1000G P3v5 data were assigned to the admix category. This revealed that in the post-QC combined dataset, there were 658 European, 76 East Asian, 46 South Asian, 51 Admixed American, 47 African, and 75 Admix individuals (Supplementary Fig. 2a). For the purpose of establishing genetic ancestry only, we also conducted a principal component analysis (PCA) of the genotypes of 31 samples that had failed genotype QC but did have DNAm data passing QC, bringing the total of women with genetic ancestry information to 677 European, 78 East Asian, 52 Admixed America, 48 African, 47 South Asian and 82 Admix individuals (Table 1).

**Analysis of DNAm data.** After QC and processing, principal component analysis (PCA) was performed using the M-values from the DNAm dataset and PCA plots were generated (Supplementary Fig. 2b, c and Supplementary Fig. 3). These plots were analyzed for any potential batch effects and differential clustering of the samples according to covariates. Covariates included the sample processing batch (batch I = 707 samples; batch II = 277 samples), genetic ancestry, cycle phase, endometriosis case:control status, site, and plate. The principal component partial R-square (PC-PR2) method was used to estimate the contribution of covariates (cycle phase, endometriosis status, institute, batch, plate, genetic ancestry) to the between-sample variability observed[105]. Furthermore, we performed correlation analysis between covariates and principal components (PCs) (Supplementary Fig. 4) to evaluate the importance of covariates in the analysis moving forward. Statistically significant associations of the top 10 PCs with batch, site and plate were identified. There was a partial association with cycle phase, and thus batch, site, plate, and cycle phase were included a priori as covariates in the downstream analyses.

*Batch correction.* In order to remove batch effects coming from covariates such as batch, institute, plate, and genetic ancestry, we used the SVA algorithm using R package *SmartSVA*[106]. First, the number of surrogate variables (SVs) was evaluated, and then the SVs were calculated while protecting contrasts for two variables —endometriosis case:control status and menstrual cycle phase at endometrial sample collection—as those were the two variables we were interested in exploring biologically. SVA was run separately for the stratified analyses.

*Single site analysis.* In order to elucidate the association of DNAm at a single DNAm with endometriosis case:control status and menstrual cycle phase, we applied a linear model using the Limma[107] R package with the significance cutoff of Benjamini Hochberg corrected p-value of 0.05. The following model was used: DNAm values ~ endometriosis case:control status + Cycle phase + SVs and interrogated via contrasts to capture case:control and cycle phase differences. All 347 women without endometriosis were included as controls. Both cases and controls were included in the cycle phase analysis. Data were visualized using heatmaps with the *Complex Heatmap*[108] package in R. Endometriosis case vs. control and cycle phase differences were also tested using a conservative mixed linear model (MLM) based approach, MLM-based omic association (MOA)[56] (Supplementary Note 1).

Cycle phase comparisons were based on aggregated cycle definitions— secretory (SE) vs. proliferative (PE) vs. menstrual, where secretory consists of early (ESE), mid (MSE) and late (LSE) secretory phases, and also on more finely categorized cycle phase definitions using the secretory sub-phases (ESE, MSE and LSE). A sensitivity analysis was carried out for a endometriosis case vs. control comparison using the 201 pathology-free controls. Given that larger genetic effects are observed in rASRM stage III–IV disease compared to stage I–II endometriosis[32], we repeated the analysis restricted to stage III–IV cases compared to pathology-free controls.

*Pathway analysis.* Pathway analysis was carried out on all significant DNAm sites (FDR < 0.05) as input into the analysis and significance cutoff on the pathway overrepresentation of FDR < 0.05. Statistically significant DNAm sites were mapped to genes using the *IlluminaHumanMethylationEPICanno.ilm10b4.hg19 annotation*[109] R package. The DNAm sites were mapped to genes if they were upstream of the gene body (TS1500, TS200, 5'UTR, 1stExon) or within the gene body. The resulting gene sets were analyzed for pathway overrepresentation using the *enrichKEGG* and *enrichPathway* functions from the *clusterProfiler* R package[110]. The over-representation analyses were also run using stratified gene mapping schemes based on the genomic features (upstream or gene body) from which DNAm sites were mapped. The analysis was carried out using the KEGG[111] and Reactome databases[112]. The top 5 statistically significant pathway gene sets were visualized using the *emapplot* function from the *enrichplot*[113] R package.

*Regional analysis.* Comparisons made in the single site analysis were repeated using regional analysis. Regional analysis was carried out with an R package *DMRCate*[114]. A nominal *p*-value cutoff of 0.05 was used for the input DNAm sites into the analysis and Fisher *p*-value of <0.1 was used to identify significant regions.

*Weighted correlation network analysis (WGCNA).* Weighted correlation network analysis (WGCNA)[33] was applied to a reduced dataset using 50% of most variable DNAm sites resulting in a dataset of 379,672 DNAm sites from the 984 samples. First, co-methylated modules were identified using the 'blockwiseModules' command[33] with minimum block size of 30 and maximum block size of 40,000 selected. Associations between the two traits of interest (endometriosis case:control status (all 347 controls) and menstrual cycle phase) and the modules were identified through use of the *Limma*[107] R package linear models using the SVs from SVA as covariates. Statistical significance was defined as a Benjamini-Hochberg adjusted *p*-value of <0.05. Significant modules were aggregated into gene sets. More specifically, for each linear model, all DNAm sites in modules with positive direction of effect were mapped to genes using the *IlluminaHumanMethylationEPICanno.ilm10b4.hg19 annotation*[109] and aggregated into a gene set, and all DNAm sites in modules with negative direction of effect were mapped to genes and aggregated into a gene set. The resulting gene sets were analyzed for pathway overrepresentation using the *enrichKEGG* and *enrichPathway* functions from the *clusterProfiler* R package[110].

**Methylation quantitative trait loci (mQTL) analysis.** DNAm and genotype data from 658 European samples was used to test for associations between genotype and DNAm in endometrium (Table 1). DNAm M-values at a total of 759,345 DNAm sites, as determined in prior analyses, and genotype data for 5,290,992 single nucleotide polymorphisms (SNPs) with imputation quality info scores > 0.8 and MAF > 0.05, were included in the analysis. Associations between genotype and methylation at DNAm sites within a *cis* distance of 1 Mb was carried out using a linear regression model in the *MatrixeQTL*[115] R package. Menstrual cycle phase, endometriosis case:control status and 39 SVA components were included as covariates in the model to adjust for known biological and technical variation. Women without endometriosis were classified as controls ($n = 221$). A Bonferroni threshold of *p*-value < $1.7 \times 10^{-11}$ was applied to account for multiple testing. Independent mQTL signals were identified using a stepwise model selection procedure in GCTA[116] for Bonferroni significant DNAm sites. Due to extensive storage requirements to compute the genomic inflation factor ($\lambda$), $\lambda$ was calculated based on associations of 759,345 DNAm sites with a random subsample of 100,000 SNPs selected from the 1000 Genomes Project panel (EUR phase 3).

*Overlap with mQTLs in other tissues.* We compared mQTLs identified in endometrium to those that have been reported as significant in blood[34,35], skeletal muscle[36], adipose tissue[37], and brain[38]. All datasets were previously generated using the Illumina HumanMethylation450 array with the exception of significant blood mQTLs reported by Hannon et al. who used the Illumina EPIC array allowing comparison of the additional DNAm sites on this array. Full summary statistics were only available for blood mQTLs published by Min et al.[46], allowing us to match *cis*-mQTLs between the tissues based on the same eSNP and DNAm site associations and direction of effect. We also limited the analysis to those DNAm sites and SNPs present in both datasets to calculate the proportion of endometrial mQTLs shared with blood. For the remaining tissues where the full SNP information was not available, we matched *cis*-mQTLs between the tissues based on DNAm site. Genes annotated to DNAm sites with mQTLs unique to endometrium were included in a pathway analysis in FUMA.

*Context-specific mQTLs.* To investigate if the effects of mQTLs differed between women with and without endometriosis and between proliferative and secretory menstrual cycle phases, we conducted context-specific mQTL analyses. A total of 118,185 independent Bonferroni significant mQTLs were included in the analysis. Anova was used to compare the following two models:

H0: DNAm ~ genotype + condition + covariates

H1: DNAm ~ genotype + condition + condition*genotype + covariates

Conditions tested included endometriosis case vs. control and proliferative (PE) versus secretory (ES + MS + LS) menstrual cycle phases. SVA components and either menstrual cycle phase or endometriosis, depending on the condition tested, were included as covariates. H1 differed from H0 by the inclusion of an interaction term between genotype and the condition being tested. To account for multiple testing, statistically significant context-specific mQTLs for each condition were defined as those with a $p$-value $< 4.2 \times 10^{-7}$ (0.05/118,185). Results were filtered to only include comparisons that had at least 10 samples homozygous for the minor allele in each group.

*Association between genetic regulation of DNAm, transcription, and endometriosis.* Summary data-based Mendelian Randomization (SMR)[39] was used to test the association between genetic variants, DNAm levels and endometriosis risk by integrating mQTL summary statistics with endometriosis GWAS summary statistics. Summary statistics used were generated from a subset of European cohorts from the largest endometriosis GWA meta-analysis to date conducted by Rahmioglu et al.[10] with the addition of a FinnGen endometriosis cohort (Supplementary Data 18). Datasets that were used in generation of the mQTL analysis (ENDOX, Melbourne, UCSF) were not included in the meta-analysis to ensure independent datasets. SMR was run for overall endometriosis (31,021 cases and 524,422 controls) and stage III/IV endometriosis (8765 cases and 373,626 controls) separately. Of note, controls in the GWA meta-analysis were population controls who did not have documentation of endometriosis diagnosis. The HEIDI (Heterogeneity In Dependent Instruments) test was also conducted to distinguish independent overlapping signals and pleiotropy/causal associations. Associations with an SMR $p$-value of $P_{SMR} < 0.05/$(number of DNAm sites tested) and a HEIDI $p$-value of $P_{HEIDI} > 0.05/$(number of genes passing the SMR test) was applied as the threshold for statistical significance.

Consistent association signals at genomic loci across multiple omics layers can help identify functionally relevant genes and regulatory elements. As such we also used SMR to test the association between mQTLs and eQTLs in endometrium[24,41], blood[117], and 49 GTEx Tissues[95] using the top associated mQTL in each dataset, and the association between eQTLs and endometriosis using the top associated eQTL. We combined results from pleiotropic associations between DNAm, gene expression and endometriosis using the stringent criterion that both the DNAm site and gene of each pair were statistically significantly associated with endometriosis at a genome-wide threshold with no association rejected by the HEIDI test. The gene–endometriosis association analysis was not dependent on the DNAm–endometriosis association analysis to account for potential discrepancies in SNPs tagging the causal SNP between the datasets.

Association between SNPs significant in the SMR analyses and other traits and diseases was investigated using PhenoScanner[118] and GWAS Catalog[119]. EpiMap[96] was also used to functionally annotate SMR significant SNPs and DNAm sites using epigenome maps from relevant tissues (uterus, ovary, vagina, mammary/breast epithelium, mammary/breast fibroblasts, skin epithelium, skin fibroblasts, Ishikawa cells, T47D cells) to define chromatin states, high-resolution enhancers, enhancer modules, upstream regulators and downstream target genes. Valid promoter-associated chromatin loops generated from H3K27Ac HiChIP libraries from a normal immortalized endometrial cell line (E6E7hTERT)[97] were also used to annotate SNPs and DNAm sites located in chromatin interaction anchor points.

**Estimation of variance captured by genetics and DNAm in endometrium.** Genome-based restricted maximum likelihood (GREML), as implemented in the Genome-wide Complex Trait Analysis (GCTA) software[116], was used to estimate the variation in endometriosis case-control status captured by common genetic variants (SNPs), also known as the SNP-based heritability. The genetic relationship matrix (GRM) used in the GREML analysis was calculated using genotype data for 5,358,309 SNPs in 953 individuals in GCTA. Similarly, omics residual maximum likelihood analyses (OREML), as implemented in the Omic-data-based Complex Trait Analysis (OSCA) software[56], was used to estimate the proportion of variance in endometriosis case-control status captured by DNAm sites and SNPs. Three different OREML models were compared, one including all DNAm sites ($n = 759,345$) for 984 individuals in the form of an omics relationship matrix (ORM), one including the ORM and the GRM for 953 individuals with matched genetic and DNAm data and one including the ORM, calculated following exclusion of DNAm sites with mQTLs ($n = 651,718$), and the GRM for the 953 individuals.

**Targeted differential DNAm analysis investigating associations with endometriosis surgically visualized and endometriosis-associated symptom sub-phenotypes.** Differential DNAm analysis for 11,698 DNAm sites within 500 Kb ± of 44 lead SNPs genome-wide statistically significantly associated with endometriosis[10] was conducted for 11 comparisons of endometriosis-case-specific surgical sub-phenotypes (rASRM stage I/II disease, stage III/IV disease; the presence of at least one superficial peritoneal lesion, endometrioma, or deep lesion; presence of at least one each of superficial peritoneal, endometrioma, and deep lesions) and six comparisons of common endometriosis-associated pain symptom sub-phenotypes (dyspareunia, acyclic pelvic pain, dyschezia) (see Supplementary Data 16 for detailed descriptions). These comparisons were restricted to NUPP controls. Only 10% of cases and NUPP controls reported no dysmenorrhea (cyclic pain with menses), and therefore we could not investigate the presence or absence of dysmenorrhea as an independent sub-phenotype. In the differential DNAm analysis, post-QC M-values from the DNAm data were utilized. Differential DNAm analysis was conducted using the Limma[107] R package. For each sub-phenotype analysis, an independent set of SVs was generated and included in the models. Both a genome-wide Bonferroni multiple testing correction ($p < 4.27 \times 10^{-6}$) and a less stringent locus-specific Bonferroni multiple testing correction (0.05/N of DNAm sites per GWAS locus) was applied. The statistically significantly differentially methylated sites were checked in the endometrium mQTL map and SMR results generated. If an mQTL was identified for a differentially methylated probe, whether the DNAm-associated SNP was in LD with an endometriosis-associated SNP in the respective region was determined. Associated DNAm sites were also annotated using EpiMap. Correlation between the effect sizes of differentially methylated sites between subtypes were calculated utilizing Spearman's rank correlations.

**Statistics and reproducibility.** All the analysis and code was implemented in R and shared via github for reproducibility. Power analysis was conducted prior to the study to ensure the findings would be robust.

**Reporting summary.** Further information on research design is available in the Nature Portfolio Reporting Summary linked to this article.

## Data availability

Methylation data used in this study has been deposited and is available from GEO (GEO: GSE223817). Genotype data generated in this study is available upon approval from dbGAP (phs003307.v1). All Supplementary Data is included in Supplementary Data 1–20.

## Code availability

Code used to run the analyses is available on github (https://github.com/SallyMortlock/Endometrial-DNA-Methylation-Endometriosis-Study). Any additional information required to reanalyze the data reported in this paper is available from the lead contact upon request.

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

## Acknowledgements

This work has been supported by the National Institutes of Health (NIH) NICHD R01 HD089511. It was also supported, in part, by funding from Wellbeing of Women (through sponsorship from PwC) (R42533) and the Medical Research Council (MR/N024524/1 and MR/N022556/1) and NIH HD094842 (Harvard/MSU). K.K. was supported by NIH NCI R37 CA233774. A.F.M. was supported by an Australian Research Council Future Fellowship (FT200100837). G.W.M. was supported by NHMRC Fellowship (GNT1177194).

## Author contributions

Concept and design: L.G., M.S., S. Mi, G.W.M., K.Z., P.A.W.R., S.H., S. Mo, C.B.; Sample procurement, clinical metadata review, sub-phenotyping: C.N., P.A.W.R., K.C.V., E.M.,

Z.G., S.S., A.J., J.L., A.W.H., J.O.A., F.C., P.S., C.B., X.S.M., J.V.J.; S.H., J.C.I., A.F.V., K.L.T., S. Mi, M.S., L.G.; Sample processing, DNA preparation, QA/QC, raw data generation: S.H.; Data QC for analysis: N.R., S.H., I.K., S.V.A., M.P., S. Mo, A.F.V., M.S.; Data analysis and interpretation: S. Mo, S.H., I.K., N.R., A.F.V., S.V.A., P.G., M.P., A.W.H., C.B., J.C.I., A.F.M., K.K., P.A.W.R., K.Z., G.W.M., S. Mi, M.S., L.G.; Human subjects assurance: K.C.V., J.C.I., L.G., C.B., P.A.W.R., A.W.H., X.S., C.N.; Drafting of the manuscript: S. Mo, S.H., I.K., N.R., K.Z., G.W.M., P.A.W.R., K.L.T., M.S., S. Mi, L.G.; All authors have read and approved the final manuscript.

## Competing interests

The authors declare no competing interests.

## Additional information

[1]The Institute for Molecular Bioscience, The University of Queensland, Brisbane, QLD 4072, Australia. [2]Center for Reproductive Sciences, Department of Obstetrics, Gynecology & Reproductive Sciences, University of California San Francisco, San Francisco, CA, USA. [3]Bakar Computational Health Sciences Institute, University of California San Francisco, San Francisco, CA, USA. [4]Wellcome Centre for Human Genetics, University of Oxford, Oxford, UK. [5]Oxford Endometriosis CaRe Centre, Nuffield Department of Women's and Reproductive Health, John Radcliffe Hospital, University of Oxford, Oxford, UK. [6]Stanford University Medical Center, Palo Alto, CA, USA. [7]University of California San Francisco, San Francisco, CA, USA. [8]Camran Nezhat Institute, Center for Special Minimally Invasive and Robotic Surgery, Woodside, CA, USA. [9]Obstetrics and Gynecology Epidemiology Center, Brigham and Women's Hospital and Harvard Medical School, Boston, MA, USA. [10]MRC Centre for Reproductive Health, University of Edinburgh, QMRI, Edinburgh, UK. [11]Carlos Simon Foundation, Health Research Institute, Valencia, Spain. [12]Group of Biomedical Research in Gynecology, Vall d'Hebron Research Institute, Barcelona, Spain. [13]Centre for Inflammation Research, Institute for Regeneration and Repair University of Edinburgh, Edinburgh, UK. [14]Department of Physiological Nursing, University of California San Francisco, San Francisco, CA, USA. [15]Department of Epidemiology, Harvard T.H. Chan School of Public Health, Boston, MA, USA. [16]Boston Center for Endometriosis, Boston Children's Hospital and Brigham and Women's Hospital, Boston, MA, USA. [17]University of Melbourne Department of Obstetrics and Gynaecology, Royal Women's Hospital, Melbourne, Australia. [18]Division of Adolescent and Young Adult Medicine, Department of Medicine, Boston Children's Hospital and Harvard Medical School, Boston, MA, USA. [19]Department of Obstetrics, Gynecology, and Reproductive Biology, College of Human Medicine, Michigan State University, Grand Rapids, MI, USA. [20]Department of Pediatrics, Division of Neonatology, University of California San Francisco, San Francisco, CA, USA. [21]These authors contributed equally: Sally Mortlock, Sahar Houshdaran, Idit Kosti, Nilufer Rahmioglu. [22]These authors jointly supervised this work: Krina Zondervan, Grant W. Montgomery, Stacey Missmer, Marina Sirota, Linda Giudice. ✉email: s.mortlock@imb.uq.edu.au; Linda.Giudice@ucsf.edu

