## [Peer Review File · Communications Biology]

Reviewers' comments:

Reviewer #1 (Remarks to the Author):

Sally Mortlock et al have performed an extensive evaluation of methylation data from endometrial tissues in relation to endometriosis risk and menstrual cycle phase. Data has been integrated with germline variant data and their data has been systematically compared with multiple resources – previous GWAS association results on related traits and outcomes, expression QTLs from GTEx and others. Findings from their efforts are certainly interesting, highlighting a significant proportion of variance in endometriosis case control status is captured by DNA methylation differences and that much of these are likely to be due to methylation at polygenic sites with individual modest effects. The results further highlight the importance of directly evaluating disease relevant tissues to identify potential dysfunctional genes and biological pathways, many of which were not picked up in data from blood and other tissues. However, I do have some clarifications and suggestions.

1) The post QC dataset from GWAS and EPIC array is slightly different - was there any QC procedures performed to identify potential 1st or 2nd degree related individuals and were these subjects excluded from the subsequent GWAS/mQTL and methylation analysis?

2) Cell-type composition is a large confounder in methylation analysis and it may be likely that endometrial tissues from different cycle phases have varied compositions of cell-types. Was there any evaluations done to quantify potential cell-types from methylation data of each sample and include these as covariates in the downstream analyses? This may be especially important as genomic inflation was high in some of the analyses ($\lambda = 1.16$). If such evaluations were not performed the authors should perhaps discuss how the present evaluations may have limited effects due to differences in cell compositions.

3) Were there any differences in methylation sites identified for these traits (for eg. cycle phases) between ethnic groups?

4) WGCNA were performed using almost half of all DNA methylation sites. Were similar pathways found the traits analyzed when using only significant methylation sites ($FDR < 0.05$)?

5) Would there be a significant enrichment for the 21 genes annotated from DNAm across menstrual cycle that also overlaps with genes from the Endometrial Receptivity Array? Was the ABCC3 gene the only gene differentially methylated between early and mid-secretory phase and was the direction of gene expression effects expected as one moves from early to mid-secretory phase?

6) While 12,242,158 significant cis-mQTLs were identified in the study, very few of these seemed to overlap with GWAS risk SNPs for endometriosis (51)? Would this suggest that the genetic control of methylation at the endometrium is largely different from the genetic control of endometriosis risks?

7) For the 31 mQTLs that were associated with the expression of 18 genes in various GTEx tissues, blood, and endometrium – was the direction of gene expression consistent with the mQTL effects (allele that increases methylation shows a decreased gene expression effect?) and were there any differences in gene expression directions when comparing data from different tissues?

8) For the findings from SMR that overlapped mQTLs with GWAS data (for eg. 10p12, 12q12 signals for ovarian cancer, endometriosis GWAS, etc) and eQTL data (eg. ADK), was there also significant colocalization of signals (for eg. using coloc analysis).

Reviewer #2 (Remarks to the Author):

This paper by Mortlock and coworkers is a comprehensive analysis of meQTL in the context of endometriosis. The study is exhaustive and based upon 679 samples analyzed compared with 201 controls.

The methylation profile was evaluated using the Illumina EPIC microarray. This is an extremely robust choice, with a lot of references in the literature. As far as I may know from the methods, the filtering and analysis were performed according to the standards in the field.

Overall, it is a nice snapshot of a complex disease in view of its methylation status. It brings a lot to the knowledge database of endometriosis, and I strongly support the accessibility of this dataset. It is very descriptive but potentially useful.

Overall, I have a few suggestions and questions, that might, in my opinion, improve the very rich messages carried by the text. The volume of information is so dense that it is almost impossible to provide a univocal message, understandably. As such, the discussion is interesting but very general for most of its content.

1. I wonder whether it would be possible to increase the size of the abstract for including some sentences from the conclusion of the paper, in particular what is written in line 557-565, which in my opinion stresses the importance of the work and of the results, and prompts the readers to go to further details.

2. It would be relevant to have the detail of the controls and patients from each source, since the DNA samples originated at least four locations worldwide (UCSF, UM, ENDOX and EDIN). If most of the controls originate from one source, this could represent a possible bias.

3. Since the analysis was performed on endometrial tissue, the authors analysed their data in regard of the moment of the cycle. It could be relevant to mention general reviews about this such as evoked recently (Retis-Resendiz, Clin Epigenetics, 2021). Also, a recent paper (Zukauskaitė et al, IJBCB) mentions the fact that DNA methylation does not seem to affect decidualization genes, but it does not mean that the expression of some other genes could be correlated with methylation differences, this study could be interesting to mention.

4. The assignment to ethnical groups is interesting. I guess it was obtained from the SNP array? It could be interesting to test the same assignation to ancestries using the methylation data. In our hands, using the same beadchip array (EPIC) than the authors, we reached the conclusion that the assignment was also doable but was in fact due not to the 'classical' CpG, but rather to those that correspond also to genetic variants (SNPs). I would be interested (and it may be of interest for the other readers as well) to know the classification of the genetic ancestry done only on these CpG-SNPs (I know that there are more than 100,000 of them in the EPIC array), and done only on the ones that can be affected by methylation differences only (pure CpGs, when the differences comes from the methylation and not from the genotype – if a CG is in some patients a AG, the methylation level will improperly be found at 0 in these patients).

5. About the confounding variables that were subtracted from the analysis, It might be interesting to have an evaluation of their weight in the total variance. In brief, a table with the part of variation coming from the batch effect, the cycle effect, the disease status (maybe in classifying the disease, and not only giving the general 15.4% value (line394)), and so on could be interesting.

Reviewer #3 (Remarks to the Author):

This manuscript by Mortlock et al. titled "Multi-omic Analysis of Global Endometrial DNA Reveals Insights into mQTL Regulation, Associated Endometriosis Disease Risk, and Endometrial Function" constitutes a comprehensive study with a large number of patients, possibly the largest, aimed at identifying differences in DNA methylation that may play a significant role in predicting endometriosis.

While the results do not provide a clear panel of DNA methylation that can differentiate between patients with endometriosis and controls, there are more apparent differences when focusing on stages III/IV of the disease. The data generated in this manuscript may not be immediately useful for predicting endometriosis in the near future, but the results related to the menstrual cycle will be a valuable resource for future investigations into endometrial biology.

However, I have some questions that need to be addressed:

MAJOR concerns:

1. The title of the manuscript states "Global Endometrial DNA Multi-omics," but the entire paper focuses on studying global methylation. I suggest changing the title to a more appropriate one based on the results.
2. Line 167, approximately two-thirds of the study participants are European, but it is unclear which ones are controls and which ones have endometriosis. Looking at the PCA, for example, Africans differ significantly from Europeans. Is there a specific pattern of methylation related to ancestry, and how do the authors correct for these differences?
3. Line 199, could the authors provide a better explanation of the biological significance of the differences in DNAm in stage III/IV cases?
4. Line 214, what are the final conclusions we can draw from the overexpressed KEGG pathways?

MINOR concerns - Typographical and grammatical errors:

- Line 563, delete "in."

We thank all the reviewers for the time taken to review our manuscript and for providing constructive feedback and comments that we have addressed to improve the quality of our manuscript.

Reviewers' comments:

Reviewer #1 (Remarks to the Author):

Sally Mortlock et al have performed an extensive evaluation of methylation data from endometrial tissues in relation to endometriosis risk and menstrual cycle phase. Data has been integrated with germline variant data and their data has been systematically compared with multiple resources – previous GWAS association results on related traits and outcomes, expression QTLs from GTEx and others. Findings from their efforts are certainly interesting, highlighting a significant proportion of variance in endometriosis case control status is captured by DNA methylation differences and that much of these are likely to be due to methylation at polygenic sites with individual modest effects. The results further highlight the importance of directly evaluating disease relevant tissues to identify potential dysfunctional genes and biological pathways, many of which were not picked up in data from blood and other tissues. However, I do have some clarifications and suggestions.

We would like to thank the reviewer for the clear summary and recognizing the value in the presented work.

1) The post QC dataset from GWAS and EPIC array is slightly different - was there any QC procedures performed to identify potential 1st or 2nd degree related individuals and were these subjects excluded from the subsequent GWAS/mQTL and methylation analysis?

We acknowledge that the post-QC datasets from the methylation and genotyping differ in sample number due to some samples not passing the stringent genotyping QC. The genotyping QC is outlined in the methods and included removal of samples with “genotype call rate < 95%, heterozygosity rate > 3 standard deviations away from the mean heterozygosity rate and removal of related (IBD > 0.200) samples” and these were removed from the subsequent GWAS/mQTL analyses. This is now clarified on page 30 (line 820) of the manuscript.

“Not all samples that passed the methylation QC also passed the genotype QC and as such only samples passing both were included in the subsequent mQTL analysis.”

2) Cell-type composition is a large confounder in methylation analysis and it may be likely that endometrial tissues from different cycle phases have varied compositions of cell-types. Was there any evaluations done to quantify potential cell-types from methylation data of each sample and include these as covariates in the downstream analyses? This may be especially important as genomic inflation was high in some of the analyses ($\lambda = 1.16$). If such evaluations were not performed the authors should perhaps discuss how the present evaluations may have limited effects due to differences in cell compositions.

We would like to thank the reviewer for bringing up this important point. We agree that cell-type composition significantly contributes to variation in methylation. While we have explored existing approaches extensively, unfortunately reference datasets for endometrial cell types are not available to accurately estimate cell-type proportions to include as covariates. We have explored alternative methods to correct for cell composition including REFRACTOR; however because reference cell profiles were developed on blood data and not endometrium this introduced noise and bias. From our work leveraging transcriptomics cell type enrichment approaches, we observed a significant difference whether the reference datasets were from blood or tissue of interest (Bunis, Wang et al, Frontiers Immunology, 2022). We have instead accepted this as a limitation and have now included this point in the discussion on page 17 (lines 466-474).

“Moreover, endometriosis case vs. control differences may be affected by cell-type composition of tissues, endometriosis disease heterogeneity or mediated by previous use of medications or behavioral changes to reduce symptoms^{59,60}. Single-cell RNA-sequencing and deconvolution of bulk endometrial samples has highlighted novel signatures for cellular subtypes in endometrium that likely contribute to variation between samples⁶¹⁻⁶³. However, in the absence of reference datasets for these cell-types appropriate corrections for cell-type composition cannot be applied to methylation analyses. Future investigations into cell type-specific effects may be facilitated by emerging technologies in single-cell epigenetics.”

3) Were there any differences in methylation sites identified for these traits (for eg. cycle phases) between ethnic groups?

We thank the reviewer for this very interesting question. We did not observe large effects of genetic ancestry on variation in methylation as shown in dimensionality plots in Supplementary Figure 3. Upon testing the correlation and association between the top 10 methylation PCs and genetic ancestry, we identified small significant correlations with PCs 3, 5, 6 and 7 (Supplementary Figure 4) the largest and most significant being with PC5. This is also confounded by the institute which is correlated with ancestry. As such, we use surrogate variable analysis to remove batch effects coming from covariates such as batch, institute, plate, and genetic ancestry in all our methylation analyses to account for any subtle effects of ethnicity. We also restrict mQTL analyses to Europeans to avoid effects of differences in LD and allele frequencies between ancestries. We do not test for differences in methylation for these traits between ethnic groups specifically as this was not a focus of this study and many of the non-European ancestral groups have very small sample sizes and lack power. Future work in more diverse populations is instrumental in answering this important question. We now include this point in the discussion on page 19 (lines 533-537).

“Interestingly, prior to correction with SVs we also observe variation in endometrial methylation associated with genetic ancestry however, given the small sample size of the non-European ancestral groups we lacked power to test for differences between ancestries. Future work in more diverse populations is instrumental in investigating ancestry dependent effects.”

4) WGCNA were performed using almost half of all DNA methylation sites. Were similar pathways found the traits analyzed when using only significant methylation sites (FDR < 0.05)?

Probes for WGCNA were chosen based on variability and co-methylation not significance from the single site analysis. Given that very few sites were significant between cases and controls in the single site analysis, no pathway analysis was conducted for those sites; however, we have now compared pathways enriched between sites significantly differentially methylated between cycle stages and those in WGCNA modules associated with cycle stage. 70% of pathways enriched for genes annotated to differentially methylated probes were also enriched in the WGCNA analysis including top pathways such as Focal adhesion, Rap1 signalling, Regulation of actin cytoskeleton and adherens junction. These pathways are now included as a supplementary table, and a statement describing the overlap has been added to the results on page 9 (lines 240-243).

“Genes annotated to probes within clusters associated with cycle phase were enriched in 330 pathways including many related to cell migration, cell-cycle progression, cytoskeleton organization, transcription, and cell proliferation (Supplementary Table 8). These pathways included 70% of the pathways identified in the single site analysis.”

5) Would there be a significant enrichment for the 21 genes annotated from DNAm across menstrual cycle that also overlaps with genes from the Endometrial Receptivity Array? Was the ABCC3 gene the only gene differentially methylated between early and mid-secretory phase and was the direction of gene expression effects expected as one moves from early to mid-secretory phase?

There was an error in this line and it should actually be 78 genes annotated to the ERA. We also tested for enrichment which was significant and have added this to the discussion on page 19. ABCC3 was the only ERA gene differentially methylated between ESE and MSE and was less methylated in MS consistent with an increased expression reported in the receptive phase (lines 523-530).

“Seventy eight genes annotated to DNAm sites differentially methylated across the menstrual cycle are present on the Endometrial Receptivity Array (ERA)⁷⁰, which contains 238 genes in total, representing a significant enrichment ($p=1.46 \times 10^{-6}$). One of these genes (ABCC3) was annotated to a DNAm site differentially methylated between early and mid-secretory phases with lower methylation in the mid-secretory phase consistent with previous reports of higher gene expression⁷⁰, suggesting that changes in DNAm may also influence endometrial receptivity to embryo implantation and/or the accuracy of estimating the window of implantation using these genes.”

6) While 12,242,158 significant cis-mQTLs were identified in the study, very few of these seemed to overlap with GWAS risk SNPs for endometriosis (51)? Would this suggest that the genetic control of methylation at the endometrium is largely different from the genetic control of endometriosis risks?

The role of genetic regulation of methylation in endometrial function remains to be determined but it is not specific to a disease. We would not expect a large proportion of mQTLs to overlap GWAS loci given the limited number of endometriosis GWAS loci identified ($n=42$). Rather we are interested in the proportion of identified endometriosis GWAS loci where we have some functional evidence for disease associated variants regulating

methylation in endometrium. We find evidence that ~40% of GWAS risk loci regulate methylation in endometrium which is more than that observed previously for eQTLs, suggesting the genetic contribution to endometriosis may have a bigger effect on methylation. We now clarify this point in the discussion on page 20 (lines 542-553).

“Epigenetic regulation in endometrium has the potential to influence vital aspects of endometrial function, female fertility, and disease. Integration with genetic data identified 118,185 independent cis-mQTLs including >3,000 potential tissue-specific mQTLs and numerous mQTLs associated with endometriosis risk. We identified 51 mQTLs, distributed in 21 genomic loci, that were significantly associated with endometriosis risk from the SMR analysis. Associations between endometriosis and QTLs at 10 loci are novel and have not previously been reported^{10,24,25,41}. These results provide evidence that endometriosis associated variants have functional effects on endometrial methylation for ~40% of endometriosis risk loci identified in the most recent endometriosis GWAS¹⁰. This is more than that observed previously for eQTLs^{24,41}, suggesting the genetic contribution to endometriosis may have a bigger effect on methylation and/or larger eQTL studies are needed in endometrial cell types to detect significant effects at these loci.”

7) For the 31 mQTLs that were associated with the expression of 18 genes in various GTEx tissues, blood, and endometrium – was the direction of gene expression consistent with the mQTL effects (allele that increases methylation shows a decreased gene expression effect?) and were there any differences in gene expression directions when comparing data from different tissues?

Supplementary table 15 shows the direction of effect of each variant on the DNAm site and gene expression. ~50% have the same direction of effect in which increased and decreased methylation is associated with increased and decreased expression respectively and the remaining 50% have opposite directions of effect. This is consistent with DNAm affecting both binding of transcription activators and repressors and similar proportions are reported by Wu et al. 2018. We also had a couple of examples of different genes affected in different tissues and the same genes affected in opposite directions in different tissues, and we have now included these examples in the discussion on page 12 (lines 313-327).

“Approximately 50% of mQTLs associated with eQTLs had the opposite direction of effect whilst the remaining had the same direction consistent with previous observations and the hypothesis that the binding affinity of both transcription factors and repressors on promoters and enhancers can be affected by DNAm⁴⁰. Examples of DNAm sites affecting different genes in different tissue (eg. CDC42 in blood, WNT4 in thyroid and LINC00339 in endometrium) and different effects on the same gene in different tissues were also observed (eg. ADK in lung vs colon).”

8) For the findings from SMR that overlapped mQTLs with GWAS data (for eg. 10p12, 12q12 signals for ovarian cancer, endometriosis GWAS, etc) and eQTL data (eg. ADK), was there also significant colocalization of signals (for eg. using coloc analysis).

We would like to thank the reviewer for this question. SMR also indirectly addresses the question of colocalization. Whilst COLOC is from the class of Bayesian methods, SMR is from the class of frequentist methods treating the null hypothesis as colocalization. Bayesian

colocalization approaches aim to identify a shared causal variant between two studies and to differentiate between distinct causal variants in LD. Similarly, Zhu et al. 2016 implemented the HEIDI test after the SMR test to further differentiate distinct causal variants in LD if the SMR test suggests significant association between two sets of summary statistics. Given both test for evidence of shared causal variants, we have chosen just to present results for SMR.

Reviewer #2 (Remarks to the Author):

This paper by Mortlock and coworkers is a comprehensive analysis of meQTL in the context of endometriosis. The study is exhaustive and based upon 679 samples analyzed compared with 201 controls.

The methylation profile was evaluated using the Illumina EPIC microarray. This is an extremely robust choice, with a lot of references in the literature. As far as I may know from the methods, the filtering and analysis were performed according to the standards in the field.

Overall, it is a nice snapshot of a complex disease in view of its methylation status. It brings a lot to the knowledge database of endometriosis, and I strongly support the accessibility of this dataset. It is very descriptive but potentially useful.

Overall, I have a few suggestions and questions, that might, in my opinion, improve the very rich messages carried by the text. The volume of information is so dense that it is almost impossible to provide a univocal message, understandably. As such, the discussion is interesting but very general for most of its content.

We would like to thank the reviewer for the clear summary and recognizing the value in the presented work. Please find the point by point responses below.

1. I wonder whether it would be possible to increase the size of the abstract for including some sentences from the conclusion of the paper, in particular what is written in line 557-565, which in my opinion stresses the importance of the work and of the results, and prompts the readers to go to further details.

Thank you, we have edited our abstract to align with comments made in the discussion to highlight the main results and encourage readers.

“Endometriosis is a leading cause of pain and infertility affecting millions of women globally. Herein, we characterize variation in DNA methylation (DNAm) and its association with menstrual cycle phase, endometriosis, and genetic variants through analysis of genotype data and methylation in endometrial samples from 984 deeply-phenotyped participants. We estimate that 15.4% of the variation in endometriosis is captured by DNAm and identify significant differences in DNAm profiles associated with stage III/IV endometriosis, endometriosis sub-phenotypes and menstrual cycle phase, including opening of the window for embryo implantation. Menstrual cycle phase was a major source of DNAm variation suggesting cellular and hormonally-driven changes across the cycle can regulate genes and pathways responsible for endometrial physiology and function. DNAm quantitative trait locus

(mQTL) analysis identified 118,185 independent cis-mQTLs including 51 associated with risk of endometriosis, highlighting candidate genes contributing to disease risk. Our work provides functional evidence for epigenetic targets contributing to endometriosis risk and pathogenesis. Data generated serve as a valuable resource for understanding tissue-specific effects of methylation on endometrial biology in health and disease.”

2. It would be relevant to have the detail of the controls and patients from each source, since the DNA samples originated at least four locations worldwide (UCSF, UM, ENDOX and EDIN). If most of the controls originate from one source, this could represent a possible bias.

We agree the source of controls between sites is important and as such we list the number and source of controls samples in Supplementary Table 1 and now specify this in the methods on page 26 (line 712). This table shows controls are split across sites and should not contribute to bias. We also include the contributing institute as a covariate in our analysis to remove any variation due to site of collection and processing which can be seen in the PCA in supplementary figure 2.

“Controls were sourced from all four recruitment sites in roughly equal proportions to avoid bias (Supplementary Table 1).”

3. Since the analysis was performed on endometrial tissue, the authors analysed their data in regard of the moment of the cycle. It could be relevant to mention general reviews about this such as evoked recently (Retis-Resendiz, Clin Epigenetics, 2021). Also, a recent paper (Zukauskaite et al, IJBCB) mentions the fact that DNA methylation does not seem to affect decidualization genes, but it does not mean that the expression of some other genes could be correlated with methylation differences, this study could be interesting to mention.

We thank the reviewer for this suggestion and have now included more discussion on this point with reference to these studies on pages 18 and 19 (lines 504-540).

“Numerous DNAm sites distributed across the genome are differentially methylated in human endometrium across the menstrual cycle, with variation likely driven by cyclicity of cell differentiation and turnover, and tissue-specific response to circulating steroid hormones^{8,72}. Differences observed in this study are consistent with those reported in smaller studies on endometrium with the addition of many more differentially methylated sites as a result of increased power^{21,25,72-75}. We validate previous observations of changes in methylation of DNA methyltransferases (eg. DNMT3a, EZH2), hormone response genes (eg. HOXA9, MYO3A) and genes associated with transcription regulation (eg. RUNX3), decidualization (eg. WNT4, ZBTB16, GREB1, HAND2, STAT5A, HOXA10) and embryo implantation (eg. MAPK14, ZMIZ1, PLXNA4)⁷². A recent study also highlighted the importance of epigenetic modifications during endometrial decidualization and whilst they did not observe methylation of decidualization-associated genes (WNT4, HAND2, STAT5A), they did report increased level of histone H4 hyperacetylation in these regions during in vitro ESC decidualization⁷⁶. Evidence suggests that changes in DNAm profiles across the menstrual cycle is a phenomenon that may be specific to endometrium and is not observed in blood^{25,72}.

Pathways enriched for genes annotated to DNAm sites differentially methylated between the proliferative and secretory phases reflect changes in endometrial function including cell

cycle progression (RHO and CDC42 GTPase cycle) and tissue remodeling (regulation of actin cytoskeleton, ECM organization). Seventy eight genes annotated to DNAm sites differentially methylated across the menstrual cycle are present on the Endometrial Receptivity Array (ERA)⁷, which contains 238 genes in total, representing a significant enrichment ($p=1.46 \times 10^{-6}$). One of these genes (ABCC3) was annotated to a DNAm site differentially methylated between early and mid-secretory phases with lower methylation in the mid-secretory phase consistent with previous reports of higher gene expression⁷, suggesting that changes in DNAm may also influence endometrial receptivity to embryo implantation and/or the accuracy of estimating the window of implantation using these genes. Indeed, a recent review highlighted changes in methylation observed during the implantation window and the association of alterations in methylation with defective receptivity and implantation failure^{72,78}. Consideration of changes in DNAm across the menstrual cycle is vital when assessing effects of regulation of molecular mechanisms on endometrial pathologies and fertility traits.”

4. The assignment to ethnical groups is interesting. I guess it was obtained from the SNP array? It could be interesting to test the same assignation to ancestries using the methylation data. In our hands, using the same beadchip array (EPIC) than the authors, we reached the conclusion that the assignment was also doable but was in fact due not to the ‘classical’ CpG, but rather to those that correspond also to genetic variants (SNPs). I would be interested (and it may be of interest for the other readers as well) to know the classification of the genetic ancestry done only on these CpG-SNPs (I know that there are more than 100,000 of them in the EPIC array), and done only on the ones that can be affected by methylation differences only (pure CpGs, when the differences comes from the methylation and not from the genotype – if a CG is in some patients a AG, the methylation level will improperly be found at 0 in these patients).

We can confirm that ancestry assignment was performed using genetic data. We agree looking at the ability of CpG-SNPs to determine ancestry would be interesting however given we had access to the genome-wide genotype data we considered this to be outside the scope of the current study. We have however deposited the data in GEO so investigators can explore this interesting question.

5. About the confounding variables that were subtracted from the analysis, It might be interesting to have an evaluation of their weight in the total variance. In brief, a table with the part of variation coming from the batch effect, the cycle effect, the disease status (maybe in classifying the disease, and not only giving the general 15.4% value (line394)), and so on could be interesting.

Thank you for raising this important point. PC-PR2 was used to quantify the amount of total variability of DNA methylation explained by technical covariates (institute, plate, batch) and sample characteristics (menstrual cycle phase, disease status, genetic ancestry). The largest contribution to the variability came from institute and cycle phase explaining 43.53% and 2.99% of overall methylation variation, respectively. Following correction for SVs institute explained only 0.53% whilst cycle phase explained 4.3%. This has now been included in the methods on page 31 (lines 846-849) and results on page 7 (lines 178-184) and as a supplementary table.

“The principal component partial R-square (PC-PR2) method was used to estimate the contribution of covariates (cycle phase, endometriosis status, institute, batch, plate, genetic ancestry) to the between-sample variability observed¹⁰⁴.”

“PC-PR2 was used to estimate the amount of total variability of DNA methylation explained by technical covariates (institute, plate, batch) and sample characteristics (menstrual cycle phase, disease status, genetic ancestry). The largest contribution to the variability came from institute, cycle phase and batch explaining 43.53%, 2.99% and 1.43% of overall methylation variation, respectively (Supplementary Table 3). Following correction using surrogate variable analysis (SVA), institute explained only 0.53% whilst cycle phase and endometriosis status, which were protected for, explained 4.30% and 0.03%, respectively.”

Covariate	Weighted partial R2 (%) before SVA	Weighted partial R2 (%) after SVA
Plate	0.94	1.21
Institute	43.53	0.53
Endometriosis	0.08	0.03
Cycle_Phase	2.99	4.30
Genetic_Ancestry	0.69	0.52
Batch	1.43	1.44
Total R2	48.36	8.56

Reviewer #3 (Remarks to the Author):

This manuscript by Mortlock et al. titled "Multi-omic Analysis of Global Endometrial DNA Reveals Insights into mQTL Regulation, Associated Endometriosis Disease Risk, and Endometrial Function" constitutes a comprehensive study with a large number of patients, possibly the largest, aimed at identifying differences in DNA methylation that may play a significant role in predicting endometriosis. While the results do not provide a clear panel of DNA methylation that can differentiate between patients with endometriosis and controls, there are more apparent differences when focusing on stages III/IV of the disease. The data generated in this manuscript may not be immediately useful for predicting endometriosis in the near future, but the results related to the menstrual cycle will be a valuable resource for future investigations into endometrial biology.

We would like to thank the reviewer for the clear summary and recognizing the value in the presented work.

However, I have some questions that need to be addressed:

MAJOR concerns:

1. The title of the manuscript states "Global Endometrial DNA Multi-omics," but the entire paper focuses on studying global methylation. I suggest changing the title to a more appropriate one based on the results.

We have changed the title to better reflect the focus of our study as suggested.

“Global Endometrial DNA Methylation Analysis Reveals Insights into mQTL Regulation and Associated Endometriosis Disease Risk and Endometrial Function”

2. Line 167, approximately two-thirds of the study participants are European, but it is unclear which ones are controls and which ones have endometriosis. Looking at the PCA, for example, Africans differ significantly from Europeans. Is there a specific pattern of methylation related to ancestry, and how do the authors correct for these differences?

We thank the reviewer for bringing up this important question. Supplementary Figure 2A is a PCA plot generated using the genotype data which shows the significance differences between Europeans and other ancestral groups using genetic data. We did not observe large effects of genetic ancestry on variation in methylation as shown in dimensionality plots in Supplementary Figure 3c. Upon testing the correlation and association between the top 10 methylation PCs and genetic ancestry, we identified small significant correlations with PCs 3, 5, 6 and 7 (Supplementary Figure 4) the largest and most significant being with PC5. This is also confounded by the institute which is correlated with ancestry. As such, we correct for genetic ancestry using the surrogate variable analysis and include these surrogate variables in all our methylation analyses to account for any subtle effects of ethnicity. We also restrict mQTL analyses to Europeans to avoid effects of differences in LD and allele frequencies between ancestries. We now include this point in the discussion on page 19 (lines 533-537).

“Interestingly, prior to correction with SVs we also observe variation in endometrial methylation associated with genetic ancestry however, given the small sample size of the non-European ancestral groups we lacked power to test for differences between ancestries. Future work in more diverse populations is instrumental to investigating ancestry dependent effects.”

3. Line 199, could the authors provide a better explanation of the biological significance of the differences in DNAm in stage III/IV cases?

We have extended discussion on the potential biological significance (lines 476-502):

“In support of effects of disease heterogeneity, we found evidence of genome-wide differences in methylation at four DNAm sites when cases were restricted to stage III/IV disease despite reduced power. Stage III/IV disease is a subset of cases based on the rASRM classification and is characterised by involvement of ovarian endometriomas and adhesions. Identification of DNAm associated with stage III/IV disease may therefore provide insights into molecular effects associated with endometriomas. Larger molecular effects in severe disease are

*consistent with genetic studies that show evidence of larger genetic effects in stage III/IV endometriosis and transcriptomic studies showing differences in endometrium of patients with stage III/IV versus stage I/II endometriosis support lower implantation and pregnancy rates in those with more advanced disease in both natural and assisted reproduction cycles⁶³. One of the genes annotated to these differentially methylated sites, *EEF2* (eukaryotic translation elongation factor 2), is downregulated in ectopic endometriosis lesions⁶⁴. Similarly, *TNPO2* (Transportin 2), a nuclear transport receptor highly expressed in the ovary, uterus, testis and brain, is a predicted target of a microRNA found to be differentially expressed between ectopic and eutopic endometrial tissue⁶⁵, and overexpressed in human ectopic endometrial epithelial cells⁶⁶. *ELAVL4* (ELAV Like RNA Binding Protein 4) encodes an RNA-binding protein with roles in post-transcriptional regulation of mRNAs and is predominantly expressed in neurons, pituitary and testis and has been associated with neurodegenerative diseases, type 2 diabetes and prognosis of endometrial adenocarcinoma^{67,68}. Interestingly, network analysis identified modules of DNAm sites associated with endometriosis case:control status. Genes annotated to these modules were enriched in WNT and MAPK signaling, adhesion and cancer pathways. Both the WNT and MAPK signaling pathways have been associated with endometriosis in previous expression studies and have been identified as potential treatment targets⁶⁹⁻⁷². Consequently, larger sample sizes will be needed to identify changes in methylation signals in disease subtypes and account for differences across the menstrual cycle. “*

4. Line 214, what are the final conclusions we can draw from the overexpressed KEGG pathways?

We attempted to provide more interpretation of these pathways in the discussion on page 18 however we want to avoid over-interpreting pathway results given they are dependent on the genesets in the databases used and we measured methylation that, whilst annotated to gene locations, does not prove definitive functional consequences on those genes.

(lines 496-502) “Interestingly, network analysis identified modules of DNAm sites associated with endometriosis case:control status. Genes annotated to these modules were enriched in WNT and MAPK signaling, adhesion and cancer pathways. Both the WNT and MAPK signaling pathways have been associated with endometriosis in previous expression studies and have been identified as potential treatment targets⁶⁸⁻⁷¹. Consequently, larger sample sizes will be needed to identify changes in methylation signals in disease subtypes and account for differences across the menstrual cycle.”

MINOR concerns - Typographical and grammatical errors:

- Line 563, delete "in."

Thank you. This has been corrected.

REVIEWERS' COMMENTS:

Reviewer #1 (Remarks to the Author):

Authors have addressed all points raised well and improved the manuscript.

Reviewer #2 (Remarks to the Author):

In this revision of their paper the authors answered to my concerns. Pertaining to the analysis of technical effects on the methylation the part played by the Institute (>43%) is striking, but indeed there are other examples in the literature where this variable is major. Another point is the fact that the methylation differences attributable to the disease are quite low (0.03-0.08%). In the other direction the variance part of endometriosis explained by methylation was of 15.4%
I still have a minor question: since almost 100% of the age can be explained by methylation, would not it be relevant to use the age of the patient when the uterus sample was taken, as an additional surrogate variable, or is it assumed that the periodic renewal of the uterus preclude the use of this parameter.

Reviewer #3 (Remarks to the Author):

The authors have appropriately addressed all concerns raised by this reviewer.

We would like to thank the reviewers for their comments which have much improved the manuscript. Please find the point by point responses for Reviewer 2:

Reviewer #2 (Remarks to the Author):

In this revision of their paper the authors answered to my concerns. Pertaining to the analysis of technical effects on the methylation the part played by the Institute (>43%) is striking, but indeed there are other examples in the literature where this variable is major. Another point is the fact that the methylation differences attributable to the disease are quite low (0.03-0.08%). In the other direction the variance part of endometriosis explained by methylation was of 15.4%.

We would like to thank the reviewer for the above comments, which we now include in the discussion section on page 17.

I still have a minor question: since almost 100% of the age can be explained by methylation, would not it be relevant to use the age of the patient when the uterus sample was taken, as an additional surrogate variable, or is it assumed that the periodic renewal of the uterus preclude the use of this parameter.

We would like to thank the reviewer for raising this question. We have done our best to match our case and control groups by age and therefore have chosen not to include age as an additional surrogate variable. While we do see a difference in our main case/control groups with respect to age The mean age for cases (N= 637) is 34.24 and mean age for controls (N=201) is 32.85 (see graph below), the magnitude of the difference is quite small. The range is also quite similar across the two groups - cases are 18-53 yo and controls are 18-50 yo. From the clinical expertise of the study team, we do believe that the turnover of the uterus likely precludes age as a factor in DNAm, but this is something that could be explored in future studies. We have commented on this on page 16 of the discussion as a future direction.